# High fat diet induces microbiota-dependent silencing of enteroendocrine cells

Lihua Ye[1,2], Olaf Mueller[1], Jennifer Bagwell[3], Michel Bagnat[3], Rodger A Liddle[2]*, John F Rawls[1,2]*

[1]Department of Molecular Genetics and Microbiology, Duke University School of Medicine, Durham, United States; [2]Division of Gastroenterology, Department of Medicine, Duke University School of Medicine, Durham, United States; [3]Department of Cell Biology, Duke University School of Medicine, Durham, United States

**Abstract** Enteroendocrine cells (EECs) are specialized sensory cells in the intestinal epithelium that sense and transduce nutrient information. Consumption of dietary fat contributes to metabolic disorders, but EEC adaptations to high fat feeding were unknown. Here, we established a new experimental system to directly investigate EEC activity in vivo using a zebrafish reporter of EEC calcium signaling. Our results reveal that high fat feeding alters EEC morphology and converts them into a nutrient insensitive state that is coupled to endoplasmic reticulum (ER) stress. We called this novel adaptation 'EEC silencing'. Gnotobiotic studies revealed that germ-free zebrafish are resistant to high fat diet induced EEC silencing. High fat feeding altered gut microbiota composition including enrichment of *Acinetobacter* bacteria, and we identified an *Acinetobacter* strain sufficient to induce EEC silencing. These results establish a new mechanism by which dietary fat and gut microbiota modulate EEC nutrient sensing and signaling.

*For correspondence:
rodger.liddle@duke.edu (RAL);
john.rawls@duke.edu (JFR)

Competing interests: The authors declare that no competing interests exist.

## Introduction

All animals derive energy from dietary nutrient ingestion. The energy harvested through digestion and absorption of dietary nutrients in the intestine is consumed by metabolic processes or stored as fat in adipose tissues. Excessive nutrient intake leads to metabolic disorders such as obesity and type 2 diabetes. To maintain energy homeostasis the animal must constantly monitor and adjust nutrient ingestion in order to balance metabolic needs with energy storage and energy intake. To accurately assess energy intake, animals evolved robust systems to monitor nutrient intake and communicate this dynamic information to the rest of the body. However, the physiological mechanisms by which animals monitor and adapt to nutrient intake remain poorly understood.

The primary sensory cells in the gut epithelium that monitor the luminal nutrient status are enteroendocrine cells (EECs) (*Furness et al., 2013*). These hormone-secreting cells are dispersed along the entire gastrointestinal tract but comprise only ~1% of gut epithelial cells (*Sternini et al., 2008*). However, collectively these cells constitute the largest, most complex endocrine network in the body. EECs synthesize and secrete hormones in response to ingested nutrients including carbohydrates, fatty acids, peptides and amino acids (*Delzenne et al., 2007*; *Moran-Ramos et al., 2012*). These nutrients directly stimulate EECs by triggering a cascade of membrane depolarization, action potential firing and voltage dependent calcium entry. Increase of intracellular calcium ($[Ca^{2+}]_i$) can trigger the fusion of hormone-containing vesicles with the cytoplasmic membrane and hormone release (*Sternini et al., 2008*). The apical surfaces of most EECs are exposed to the gut lumen allowing them to detect ingested luminal contents (*Gribble and Reimann, 2016*). However, some EECs are not open to the gut lumen and reside close to the basal lamina (*Höfer et al., 1999*;

*Sternini et al., 2008*). These different morphological types are classified as 'open' or 'closed' EECs respectively, and traditionally have been thought to reflect distinct developmental cell fates. However, the transition between open and closed EEC types has not been described.

Besides morphological characterization, EECs are commonly classified by the hormones they express. More than 15 different hormones have been identified in EECs which exert broad physiological effects on gut motility, satiation, food digestion, nutrient absorption, insulin sensitivity, and energy storage (*Moran-Ramos et al., 2012*). EECs communicate not only through circulating hormones, but also through direct paracrine and neuronal signaling to multiple systems including the intrinsic and extrinsic nervous system, pancreas, liver and adipose tissue (*Bohórquez et al., 2015*; *Gribble and Reimann, 2016*; *Kaelberer et al., 2018*; *Latorre et al., 2016*). EECs therefore have a key role in regulating energy homeostasis and represent the first link that connects dietary nutrient status to systemic metabolic processes.

Energy homeostasis can be influenced by many environmental factors, although diet plays the most important role. Despite efforts to reduce dietary fat intake in recent decades, the percentage of energy intake from fat remains ~33% in the US (*Austin et al., 2011*). High levels of dietary fat have a dominant effect on energy intake and adiposity (*Zhao et al., 2018*) and have been implicated in the high prevalence of human metabolic disorders worldwide (*Ludwig et al., 2018*; *Oakes et al., 1997*; *Panchal et al., 2011*). The effects of a high fat diet on peripheral tissues like pancreatic islets, liver and adipose tissue have been studied extensively (*Green and Hodson, 2014*; *Kahn et al., 2006*). It is also well appreciated that consumption of a high fat diet affects the microbial communities residing in the intestine, commonly refered to as the gut microbiota (*David et al., 2014*; *Hildebrandt et al., 2009*; *Murphy et al., 2010*; *Turnbaugh et al., 2008*; *Wong et al., 2015*). Gnotobiotic animal studies also demonstrated that gut microbiota altered by high fat diet can promote adiposity and insulin resistance (*Ridaura et al., 2013*; *Turnbaugh et al., 2008*; *Turnbaugh et al., 2006*), but the underlying mechanisms are incompletely understood. Notably, despite the importance of EECs in nutrient monitoring and systemic metabolic regulation, it remains unknown how a high fat diet might impact EEC function and whether the gut microbiota play a role in this process.

A major problem in studying the effects of diet on EEC physiology has been the lack of in vivo techniques for studying these rare cells in an intact animal. Historically, in vivo EEC function has been studied by measuring hormone levels in blood following luminal nutrient stimulation (*Goldspink et al., 2018*). However, many gastrointestinal hormones have very short half-lives and peripheral plasma hormone levels do not mirror real-time EEC function (*Cuenco et al., 2017*; *Druce et al., 2009*; *Kieffer et al., 1995*). EEC function has been measured in vitro via cell and organoid culture models using electrophysiological cellular recordings and fluorescence-based calcium imaging (*Kaelberer et al., 2018*; *Kay et al., 1986*; *Reimann et al., 2008*). However, these in vitro models are not suited for modeling the effect of diet and microbiota on EEC function as they are unable to reproduce the complex in vivo environment that involves signals from neighboring cells like enterocytes, enteric nerves, blood vessels and immune cells. Moreover, in vitro culture systems are unable to mimic the dynamic and complex luminal environment that contains food and microbiota. Therefore, to fully understand the effects of diet and microbiota on EEC function, it is necessary to study EECs in vivo.

In this study, we utilized the zebrafish model to investigate the impact of dietary nutrients and microbiota on EEC function. The development and physiology of the zebrafish digestive tract are similar to those of mammals (*Wallace et al., 2005*; *Wallace and Pack, 2003*). Zebrafish hatch from their protective chorions at 3 days post-fertilization (dpf) and microbial colonization of the intestinal lumen begins shortly thereafter (*Rawls et al., 2007*). The zebrafish intestine becomes completely patent by 4 dpf and feeding and digestion begin around 5 dpf. The zebrafish intestine develops most of the same differentiated epithelial cell types as observed in mammals, including absorptive enterocytes, mucus-secreting goblet cells, and EECs (*Ng et al., 2005*; *Wallace et al., 2005*; *Wallace and Pack, 2003*). Absorption of dietary fat occur primarily in enterocytes within the proximal intestine of the zebrafish (*Quinlivan and Farber, 2017*) (yellow area in *Figure 1D*). These conserved aspects of intestinal epithelial anatomy and physiology are associated with a conserved transcriptional regulatory program shared between zebrafish and mammals (*Lickwar et al., 2017*). The zebrafish intestine is colonized by a complex microbiota which promotes intestinal absorption of dietary fat (*Semova et al., 2012*) but microbial and nutritional effects on zebrafish EEC physiology were unknown. To monitor EEC activity in zebrafish, we used a genetically encoded calcium indicator

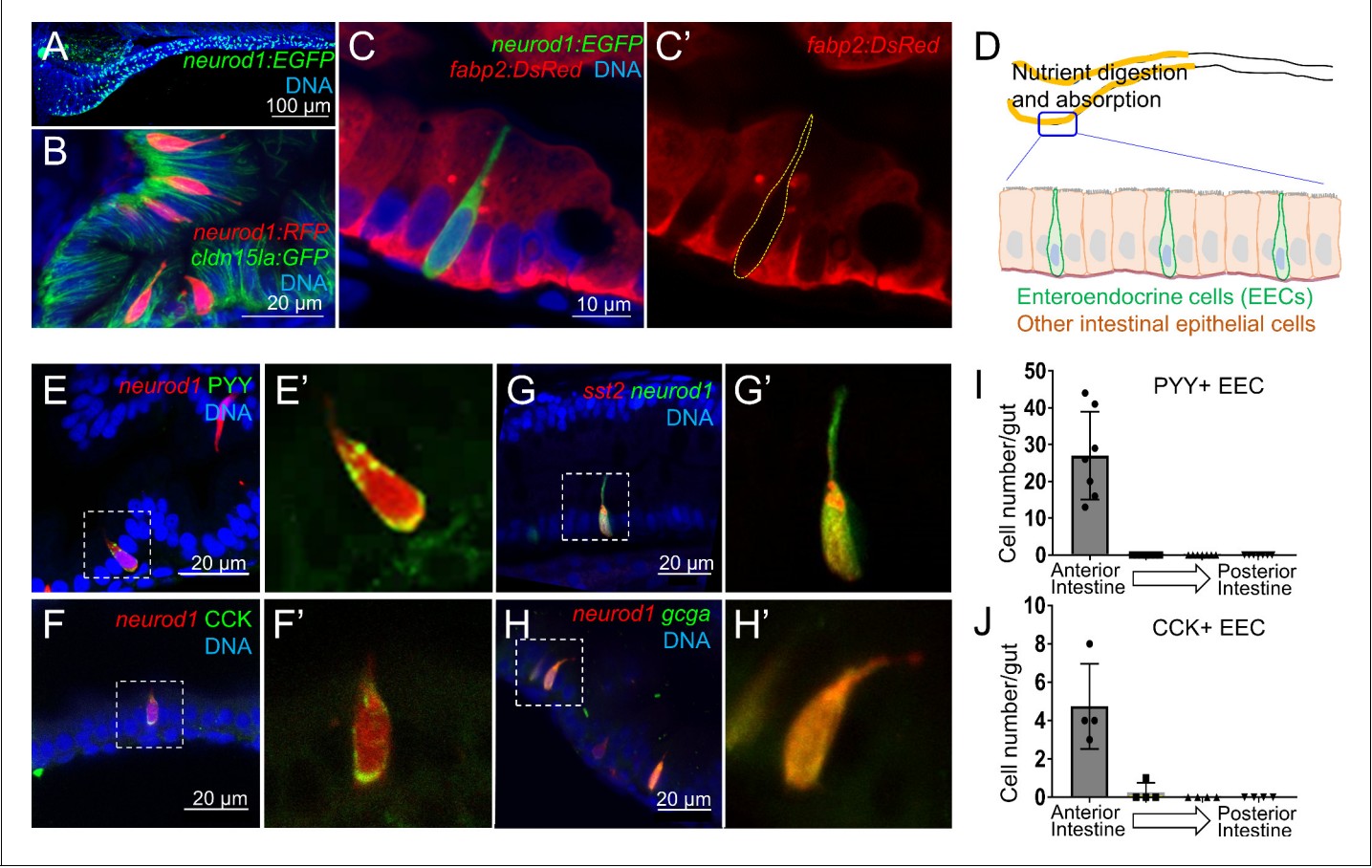

**Figure 1.** Identification of *neurod1+* enteroendocrine cells (EECs) in zebrafish. (**A**) Confocal projection of zebrafish EECs marked by the *TgBAC (neurod1:EGFP)* transgenic line. (**B**) Confocal projection of zebrafish EECs marked by *Tg(neurod1:RFP)*. *TgBAC(cldn15la:GFP)* marks intestinal epithelial cells. (**C**) Confocal image of zebrafish EECs marked by *TgBAC(neurod1:EGFP)* transgenic line. (**C'**) Subpanel image of zebrafish enterocyte marked by *Tg(fabp2:DsRed)*. Note that *neurod1+* EECs do not express the enterocyte marker *fabp2*. (**D**) Schematic diagram of 6 dpf larval zebrafish intestine. The anterior region of the intestine that is largely responsible for nutrient absorption is highlighted in yellow. (**E–F**) Confocal image of *neurod1+* EECs stained for PYY (E,) and CCK (F). (**E'–F'**) Zoom view of PYY and CCK positive EECs. (**G–H**) Confocal image of *neurod1+* EECs expressing somatostatin [marked by *Tg(sst2:DsRed)* in G] and proglucagon hormones [marked by *Tg(gcga:EGFP)* in H]. (**G'–H'**) Zoom view of *sst2* and *gcga* positive EECs. (**I–J**) Quantification of PYY+ (n = 7) and CCK+ (n = 4) EECs in 6 dpf zebrafish intestines.

The online version of this article includes the following figure supplement(s) for figure 1:

**Figure supplement 1.** Characterization of zebrafish enteroendocrine cells.
**Figure supplement 2.** Analysis of EEC lifespan in zebrafish larvae using single dose EdU labeling.

(Gcamp6f) expressed under control of an EEC gene promoter. The excitability of EECs upon luminal stimulation could be measured using in vivo fluorescence-based calcium imaging. By combining this in vivo EEC activity assay with diet and gnotobiotic manipulations, we show here that specific members of the intestinal microbiota mediate a novel physiologic adaption of EECs to high fat diet.

## Results

### Establishing methods to study enteroendocrine cell function using an in vivo zebrafish model

We first developed an approach to identify and visualize zebrafish EECs in vivo. Previous mouse studies have shown that the transcription factor NeuroD1 plays an essential role to restrict intestinal progenitor cells to an EEC fate (*Li et al., 2011*; *Ray and Leiter, 2007*), and is expressed in almost all EECs without expression in other intestinal epithelial cell lineages (*Li et al., 2012*; *Ray et al., 2014*).

We used transgenic zebrafish lines expressing fluorescent proteins under control of regulatory sequences from the zebrafish *neurod1* gene, *Tg(neurod1:RFP)* (*McGraw et al., 2012*) and *TgBAC (neurod1:EGFP)* (*Trapani et al., 2009*). We found that both lines labeled cells in the intestinal epithelium of 6 dpf zebrafish (*Figure 1A–B*, *Figure 1—figure supplement 1A*), and that these *neurod1+* cells do not overlap with goblet cells and express the intestinal secretory cell marker 2F11 (*Crosnier et al., 2005*) (*Figure 1—figure supplement 1C–E*). To further test whether these *neurod1 +* cells in the intestine label secretory but not absorptive cell lineages, we crossed *Tg(neurod1:RFP)* with the Notch reporter line *Tg(tp1:EGFP)* (*Parsons et al., 2009*). Activation of Notch signaling is essential to restrict intestinal progenitor cells to an absorptive cell fate (*Crosnier et al., 2005*; *Li et al., 2012*), suggesting *tp1+* cells may represent enterocyte progenitors. In accord, we found that *neurod1+* cells in the intestine do not overlap with *tp1+* cells (*Figure 1—figure supplement 1B*). Additionally, our results demonstrated that *neurod1+* cells in the intestine do not overlap with the mature enterocyte marker *ifabp/fabp2* (*Kanther et al., 2011*) (*Figure 1C*). These results suggested that, similar to mammals, *neurod1* expression in the zebrafish intestine occurs specifically in EECs. In addition, using EdU labeling in 5 dpf zebrafish larvae, we found that EECs in the intestine are post-mitotic and require about 30 hr to differentiate from proliferating progenitors (*Figure 1— figure supplement 2*).

Hormone expression is a defining feature of EECs, so we next evaluated the expression of four hormones in *neurod1+* EECs in 6 dpf zebrafish larvae: peptide YY (PYY), cholecystokinin (CCK), somatostatin (*Tg(sst2:RFP)*, *Li et al., 2009*) and glucagon (precursor to glucagon-like peptides GLP-1 and GLP-2; *Tg(gcga:EGFP)*, *Ye et al., 2015*) (*Figure 1E–H*). We found that PYY and CCK hormones, which are important for regulating fat digestion and feeding behavior, are only expressed in EECs in the proximal intestine where dietary fats and other nutrients are digested and absorbed (*Carten et al., 2011*; *Farber et al., 2001*) (*Figure 1I–J*). In contrast, somatostatin expression occurred in EECs along the whole intestine and glucagon expressing EECs were found along the proximal and mid-intestine (segment 2) but excluded from the posterior intestine (segment 3) (*Figure 1—figure supplement 1F–G*). The regionalization of EEC hormone expression may reflect the functional difference of EECs and other epithelial cell types along the zebrafish intestine (*Lickwar et al., 2017*).

EECs are specialized sensory cells in the intestinal epithelium that can sense nutrient stimuli derived from the diet such as glucose, amino acids and fatty acids. Upon receptor-mediated nutrient simulation, EECs undergo membrane depolarization that results in transient increases in intracellular calcium that in turn induce release of hormones or neurotransmitters (*Goldspink et al., 2018*). Therefore, the transient increase in intracellular calcium concentration is an important mediator and indicator of EEC function. To investigate EEC function in zebrafish, we utilized *Tg(neurod1:Gcamp6f)* transgenic zebrafish (*Rupprecht et al., 2016*), in which the calcium-dependent fluorescent protein *Gcamp6f* is expressed in EECs under control of the −5 kb *neurod1* promoter (*McGraw et al., 2012*). Using this transgenic line, we established an in vivo EEC activity assay system which permitted us to investigate the temporal and spatial activity of EECs in vivo. Briefly, unanesthetized *Tg(neurod1:Gcamp6f)* zebrafish larvae were positioned under a microscope objective and a solution containing a stimulus was delivered onto their mouth. The stimulus was then taken up into the intestinal lumen and EEC Gcamp6f activity was recorded simultaneously (*Figure 2A*; see Materials and methods and *Figure 2—figure supplement 1* for further details). Using this EEC activity assay, we first tested if zebrafish EECs were activated by fatty acids. We found that palmitate, but not the BSA vehicle control, activated a subset of EECs (*Figure 2B–F*, *Video 1*). Similar patterns of EEC activation in the proximal intestine were induced by the fatty acids linoleate and dodecanoate; whereas, the short chain fatty acid butyrate did not induce EEC activity (*Figure 2D*). The ability of EECs in the proximal intestine to respond to fatty acid stimulation is interesting because that region is the site of dietary fatty acid absorption (*Carten et al., 2011*). In this region EECs express CCK which regulates lipase and bile secretion and PYY which regulates food intake (*Figure 1I and J*). Our results further establish that activation by fatty acids is a conserved trait in zebrafish and mammalian EECs.

## High fat feeding impairs enteroendocrine cell nutrient sensing

The vast majority of previous studies on EECs in all vertebrates has focused on acute stimulation with dietary nutrients including fatty acids. In contrast, we have very little information on the

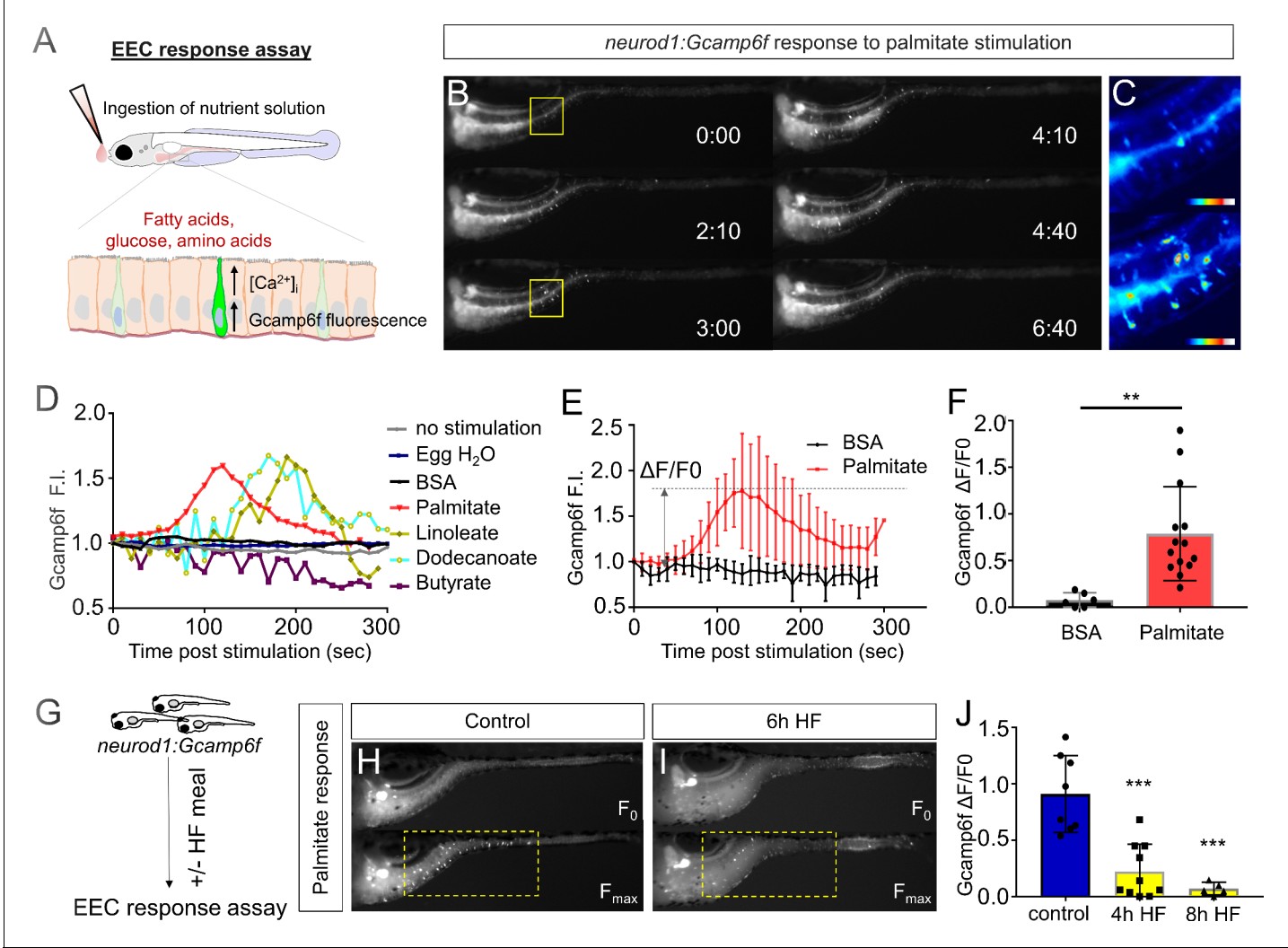

**Figure 2.** High fat feeding impairs the EEC calcium response toward palmitate stimulation. (A) Measurement of the EEC response to nutrient stimulation using *Tg(neurod1:Gcamp6f)*. (B) Time lapse image of the EEC response to BSA conjugated palmitate stimulation in *Tg(neurod1:Gcamp6f)* using the EEC response assay. Note that palmitate responsive EECs are primarily in the proximal intestine. (C) Heat map image indicating the EEC calcium response at 0 and 3 min post palmitate stimulation from the highlighted area in B. (D) Change in Gcamp6f relative fluorescence intensity in 5 min with no stimulation or stimulation with egg water, BSA vehicle, palmitate, linoleate, dodecanoate or butyrate. Note that only palmitate, linoleate and dodecanoate induced EEC calcium responses. (E, F) Change in Gcamp6f relative fluorescence intensity in BSA stimulated (n = 4) and palmitate stimulated animals (n = 5). (G) Measurement of EEC calcium responses to palmitate stimulation following 4–8 hr of high fat (HF) meal feeding in 6 dpf *Tg(neurod1:Gcamp6f)* larvae. (H, I) Representative images of the EEC response to palmitate stimulation in control larvae (without HF meal feeding, (H) and 6 hr of HF feeding (I). (J) Measurement of EEC calcium responses to palmitate stimulation in *Tg(neurod1:Gcamp6f)* larvae following 4 and 8 hr HF feeding. Student t-test was used in F and one-way ANOVA with post-hoc Tukey test was used in J. **p<0.01, ***p<0.001.

The online version of this article includes the following figure supplement(s) for figure 2:

**Figure supplement 1.** EEC activity assay.

**Figure supplement 2.** Feeding a high fat meal did not impair subsequent fatty acid intake.

adaptations that EECs undergo during the postprandial process. To address this gap in knowledge, we applied an established model for high fat meal feeding in zebrafish (*Carten et al., 2011*; *Semova et al., 2012*). In this high fat (HF) meal model, zebrafish larvae are immersed in a solution containing an emulsion of chicken egg yolk liposomes which they ingest for a designated amount of time prior to postprandial analysis using our EEC activity assay (*Figure 2G*). Importantly, ingestion of a HF meal d not prevent subsequent nutrient stimuli such as fatty acids to be ingested and distributed along the length of the intestine (*Figure 2—figure supplement 2*). To our surprise, we found

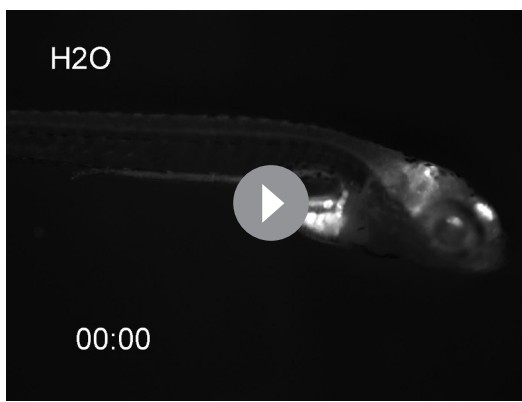

**Video 1.** EEC calcium response to water, BSA, palmitate, glucose and cysteine administration in *Tg (neurod1:Gcamp6f)* zebrafish larvae at 6dpf [10 s/frame for 5 or 10 min (cysteine)].
https://elifesciences.org/articles/48479#video1

that the ability of EECs in the proximal intestine to respond to palmitate stimulation in our EEC activity assay was quickly and significantly reduced after 6 hr of HF meal feeding (*Figure 2H–J*, *Video 2*).

We next sought to test if HF feeding only impairs EEC sensitivity to fatty acids or if there are broader impacts on EEC nutrient sensitivity. First, we investigated EEC responses to glucose stimulation. Similar to fatty acids, glucose stimulation activated EECs only in the proximal intestine of the zebrafish under unfed control conditions (*Figure 3A and B*, *Video 1*). Previous mammalian cell culture studies reported that glucose-stimulated elevation of intracellular calcium concentrations and hormone secretion in EECs is dependent upon the EEC sodium dependent glucose cotransporter 1 (Sglt1), an apical membrane protein that is expressed in small intestine and renal tubules and actively transports glucose and galactose into cells (*Song et al., 2016*). Similarly, we found that Sglt1 is expressed on the apical surface of zebrafish intestinal epithelial cells including enterocytes and EECs (*Figure 3E*). In addition, co-stimulation with glucose and phlorizin, a chemical inhibitor of Sglt1, blocked the EEC activation induced by glucose (*Figure 3F–G*). Consistently, the EEC response to glucose stimulation was significantly increased by the addition of NaCl in the stimulant solution which will facilate sodium gradient dependent glucose transport by Sglt1 (*Figure 3C*). In addition, zebrafish EECs also responded to the other Sglt1 substrate, galactose, but not fructose (*Figure 3D*). These results suggest that glucose can induce EEC activity in a Sglt1 dependent manner in the zebrafish intestine.

We then examined if HF feeding impaired subsequent EEC responses to glucose, as we had observed for fatty acids (*Figure 2G–J*). Indeed, HF feeding significantly reduced EECs' response to subsequent glucose stimulation (*Figure 3H–J*, *Video 3*). We extended these studies to investigate zebrafish EEC responses to amino acids. Among the twenty major amino acids we tested, we only observed significant EEC activity in response to cysteine stimulation under control conditions (*Figure 3—figure supplement 1A–B*, *Video 1*). However, in contrast to the fatty acid and glucose responses, zebrafish EECs that responded to cysteine were located primarily in the mid intestine (*Figure 3—figure supplement 1A–B*) and HF meal ingestion did not significantly impair subsequent EEC responses to cysteine (*Figure 3—figure supplement 1C–E*). These results collectively indicate that HF feeding impairs the function of palmitate and glucose responsive EECs in the proximal intestine, the region where fat absorption takes place.

## High fat feeding induces morphological adaption in enteroendocrine cells

To further investigate how HF feeding impacts zebrafish EECs, we leveraged the transparency of the zebrafish to permit morphologic analysis of EECs. In zebrafish under control conditions, most EECs are in an open-type morphology (*Figure 1B–G*) with an apical process that extends to the intestinal lumen, allowing them to directly interact with the contents of the intestinal lumen (*Figure 4A*). When we examined the proximal zebrafish intestine after 6 hr of HF feeding, we discovered that most EECs had adopted a

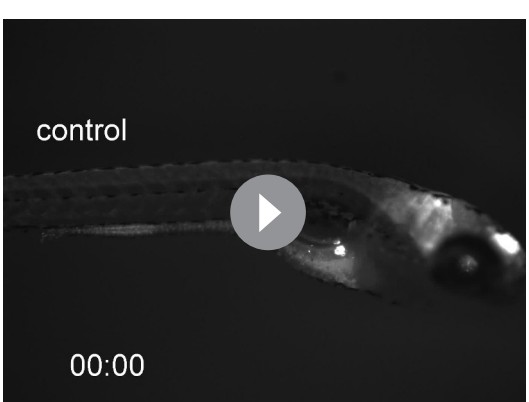

**Video 2.** EEC calcium response to palmitate stimulation in control and 6 hr high fat fed *Tg(neurod1:Gcamp6f)* zebrafish larvae at 6dpf (10 s/frame for 5 min).
https://elifesciences.org/articles/48479#video2

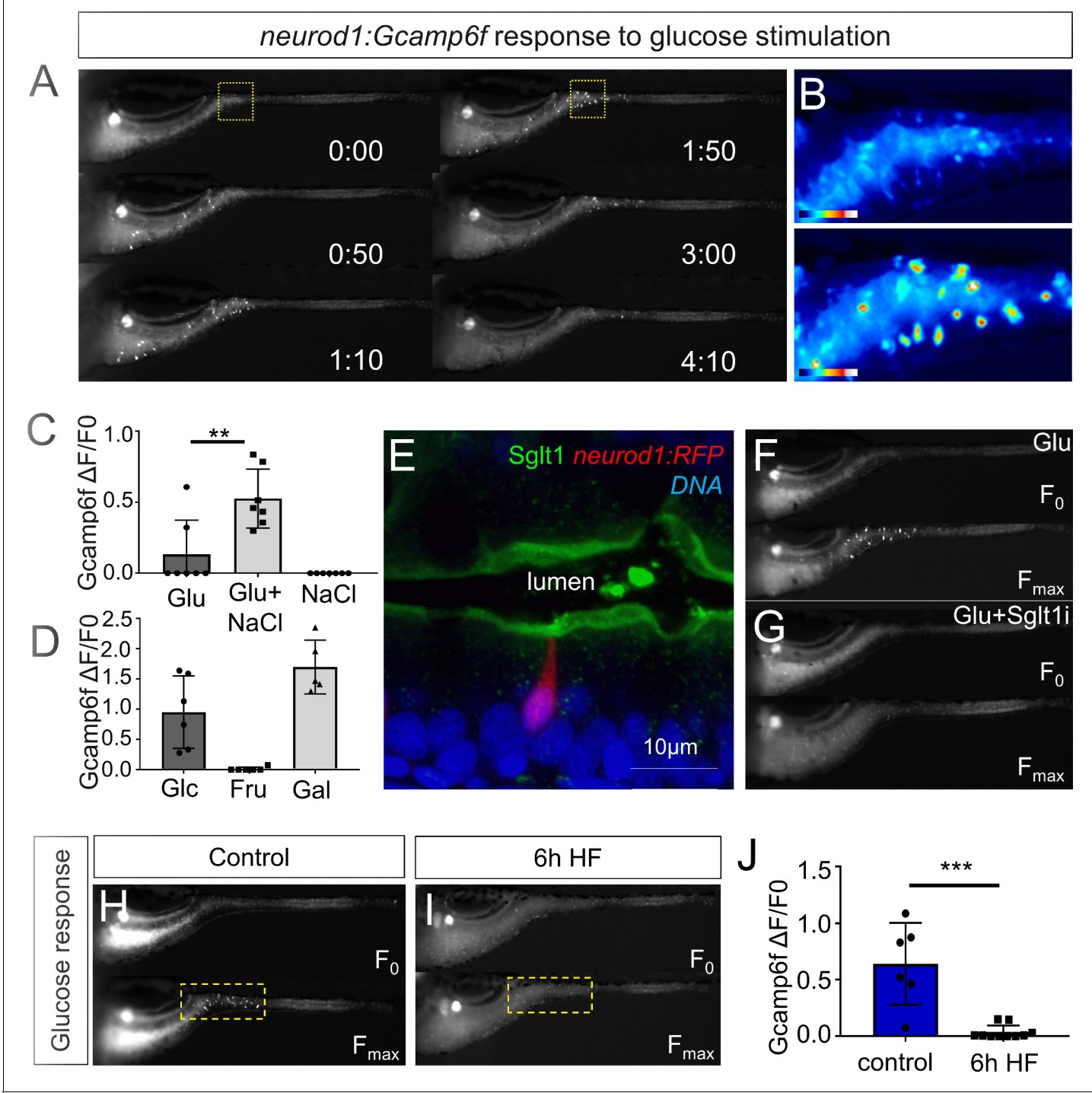

**Figure 3.** High fat feeding impairs EEC calcium response to glucose stimulation. (**A**) Time lapse images of the EEC response to glucose (500 mM, dissolved in 100 mM NaCl solution) in 6 dpf *Tg(neurod1:Gcamp6f)* larvae using the EEC response assay. (**B**) Heat map image indicating the EEC calcium response at 0 and 1min 50s post glucose stimulation from the highlighted area in A. (**C**) Measurement of the EEC calcium response when stimulated with glucose (500 mM) dissolved in water or 100 mM NaCl vehicle. Note that the presence of NaCl significantly increased the glucose induced EEC calcium response. (**D**) Measurement of the EEC calcium response when stimulated with glucose (500 mM), fructose (500 mM) and galactose (500 mM). All of these stimulants were dissolved in 100 mM NaCl vehicle. Note that only glucose and galactose induced the EEC calcium response. (**E**) Confocal image of 6 dpf zebrafish intestine stained with Sglt1 antibody. EECs were marked by *Tg(neurod1:RFP)*. Note that Sglt1 is located on the apical side of intestinal cells. (**F, G**) Representative image of the EEC calcium response in *Tg(neurod1:Gcamp6f)* when stimulated with 500 mM glucose or 500 mM glucose with a Sglt1 inhibitor (0.15 mM phloridzin). Note in G that when co-stimulated with glucose and Sglt1 inhibitor, the intestine

*Figure 3 continued on next page*

*Figure 3 continued*

appeared to dilate but no EEC activation was observed. (H,I) Representative image of the EEC calcium response to glucose stimulation in control larvae without high fat (HF) meal feeding (H) and 6 hr HF fed larvae (I). (J) Quantification of the EEC calcium response to glucose stimulation in control and 6 hr HF fed larvae. Student t-test was used in C,J. **p<0.01, ***p<0.001.

The online version of this article includes the following figure supplement(s) for figure 3:

**Figure supplement 1.** EECs remain responsive to cysteine following high fat feeding.

closed-type morphology that apparently lacked an apical extension and no longer had access to the lumenal contents (*Figure 4B*, *Figure 4—figure supplement 1A–C*). We first speculated this shift from open-type to closed-type EEC morphology may be due to cell turnover and loss of open-type EECs and replacement with newly differentiated closed-type EECs. To test this possibility, we created a new *Tg(neurod1:Gal4); Tg(UAS:Kaede)* photoconversion tracing system in which UV light can be used to convert the Kaede protein expressed in EECs from green to red emission (*Figure 4—figure supplement 2A–C*). This allowed us to label all existing differentiated *neurod1+* EECs by UV light photoconversion immediately before HF feeding (*Figure 4—figure supplement 2G*), so that pre-existing EECs emit red and green Kaede fluorescence and any newly differentiated EECs emit only green Kaede fluorescence (*Figure 4—figure supplement 2D–E*). However, we did not observe the presence of any green EECs following HF feeding (*Figure 4—figure supplement 2F–G*). To test whether HF feeding induced EEC apoptosis, we used an in vivo apoptosis assay in which *Tg(ubb:sec5A-tdTomato)* zebrafish (*Espenschied et al., 2019*) were crossed with *TgBAC(neurod1:EGFP)* allowing us to determine if apoptosis occurred in EECs (*Figure 4—figure supplement 3A–B*). However, we did not detect activation of apoptosis in closed-type EECs following HF diet feeding (*Figure 4—figure supplement 3C*). Change of cell volume is an indicator of apoptotic cells. Consistently, we also did not observe a significant change in EEC cell volume following HF feeding (*Figure 4—figure supplement 3D*). These results suggest that the striking change in EEC morphology during HF feeding is not due to EEC turnover nor EEC apoptosis but is instead due to adaptation of the existing EECs.

To analyze this adaptation of EEC morphology in greater detail, we used a new transgenic line *TgBAC(gata5:lifeAct-EGFP)* together with the *Tg(neurod1:RFP)* line. In these animals, the apical surface of EECs and other intestinal epithelial cells can be labeled by *gata5:lifeAct-EGFP* and the cytoplasmic extension of EECs to the apical lumen can be visualized and quantified through z-stack confocal imaging of the proximal intestine (*Video 4*). We measured the ratio of EECs with apical extensions to the total number of EECs, and

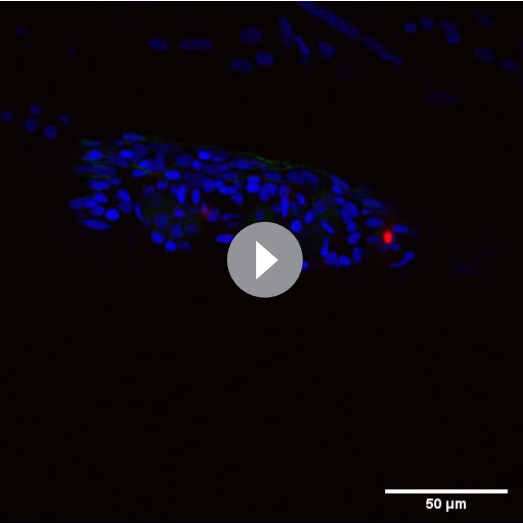

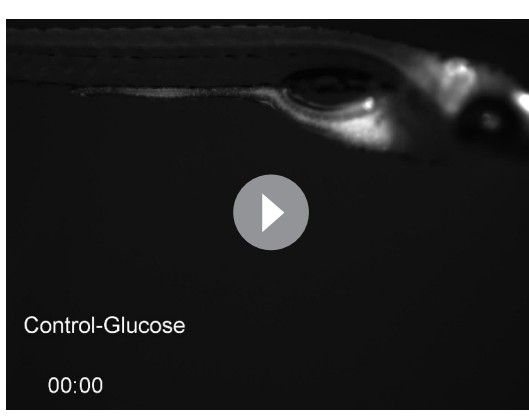

**Video 3.** EEC calcium response to glucose stimulation in control and 6 hr high fat fed *Tg(neurod1:Gcamp6f)* zebrafish larvae at 6dpf (10 s/frame for 5 min).
https://elifesciences.org/articles/48479#video3

**Video 4.** Confocal Z stack images in *Tg(neurod1:RFP); TgBAC(gata5:lifeAct-EGFP)* control zebrafish larvae at 6 dpf. The apical surface of the intestinal epithelium was labeled by *gata5:lifeAct-EGFP*. Note that the apical protrusion of EECs extend to the intestinal lumen.
https://elifesciences.org/articles/48479#video4

defined that ratio as an 'EEC morphology score'. In control larvae, most EECs are open-type and the morphology score is near 1 (*Figure 4E*). We found that the EEC morphology score gradually decreased upon HFfeeding (*Figure 4E*, *Video 5*), indicating that EECs had changed from an open-type to closed-type morphology. To further analyze the dynamics of the EEC apical response, we generated a new transgenic line *Tg(neurod1:lifeAct-EGFP)*(*Figure 4—figure supplement 4A and B*). Using in vivo confocal time-lapse imaging in *Tg(neurod1:lifeAct-EGFP)* zebrafish, we confirmed that EEC apical processes undergo dynamic retraction after HF feeding (*Figure 4F*), which was not observed in control animals (*Figure 4—figure supplement 4C and D*, *Video 6*). Interestingly, EECs in the distal-intestine retained their open-type morphology following HF feeding (*Figure 4—figure supplement 1F–H*), suggesting the adaptation from open- to closed-type EEC morphology is a specific response of EECs in the proximal intestine. This suggests that this EEC morphological adaption upon HF feeding is associated with impairment of EEC sensitivity to subsequent exposure to nutrients such as palmitate and glucose.

To investigate whether the diet-induced EEC morphology change is conserved in adult zebrafish, we performed a similar HF feeding paradigm in 1.5 year old *Tg(neurod1:RFP)* adult zebrafish and examined EEC morphology in whole mount zebrafish intestines. Our results demonstrated that, consistent with our observations in larvae zebrafish, 10 hr HF feeding triggered a similar open- to closed-type change in EEC morphology in adult zebrafish proximal intestine (*Figure 4—figure supplement 5*). This suggests that this diet-induced EEC adaptation is not restricted to larval stage animals but is a general postprandial physiological response. Next, we aimed to understand whether HF feeding-induced EEC functional and morphological adaptation is reversible. We performed similar HF feeding in *Tg(neurod1:Gcamp6f)* zebrafish larvae and transferred HF-fed zebrafish to fresh egg water for recovery. We observed that EECs' calcium response to palmitate paralleled the clearance of the HF meal from the intestine (*Figure 5A,B and G*). Twenty hours after HF feeding, intestinal fat was almost completely cleared from the intestine and the EEC calcium response to palmitate was restored comparable to that of unfed controls (*Figure 5B and H*). HF feeding-induced changes in EEC morphology was also reversible. After 20 hr of recovery from HF feeding, the apical extension of most EECs had returned to the intestinal lumen and the EEC morphology score normalized (*Figure 5C–F and I*). To investigate whether restoration of these functional and morphological features was due to recovery of existing EECs or new EEC neogenesis, we performed similar Kaede photoconvertable EEC cell tracing using the *Tg(neurod1:Gal4); Tg(UAS:Kaede)* system (*Figure 4—figure supplement 2*). The existing EECs were labeled with UV after 6 hr of HF feeding.

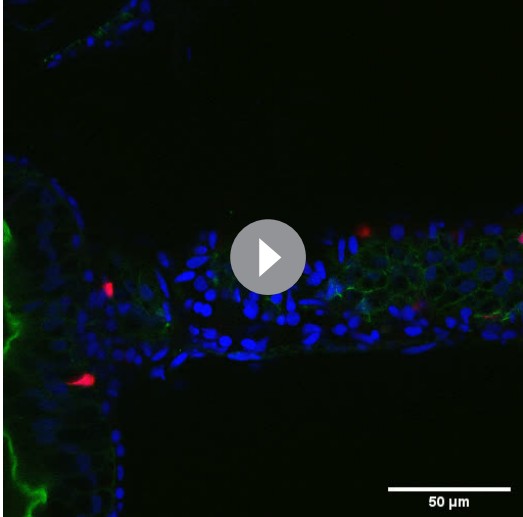

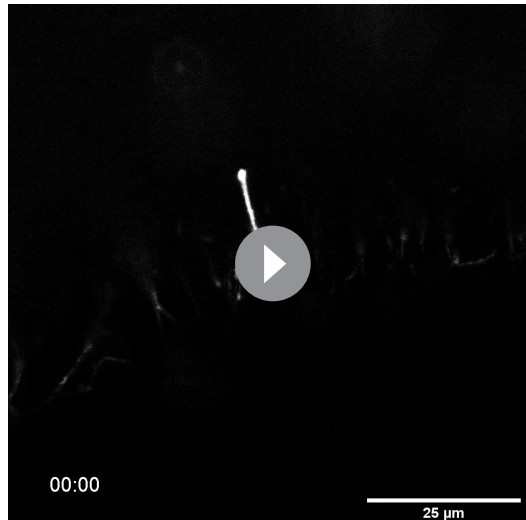

**Video 5.** Confocal Z stack image in *Tg(neurod1:RFP); TgBAC(gata5:lifeAct-EGFP)* 10 hr high fat fed zebrafish larvae at 6 dpf. Note that the majority of EECs have lost their apical protrusions.

https://elifesciences.org/articles/48479#video5

**Video 6.** Time lapse video of intestine in control *Tg (neurod1:lifeAct-EGFP)* zebrafish larvae at 6dpf. (10 s/ frame for 16 min).

https://elifesciences.org/articles/48479#video6

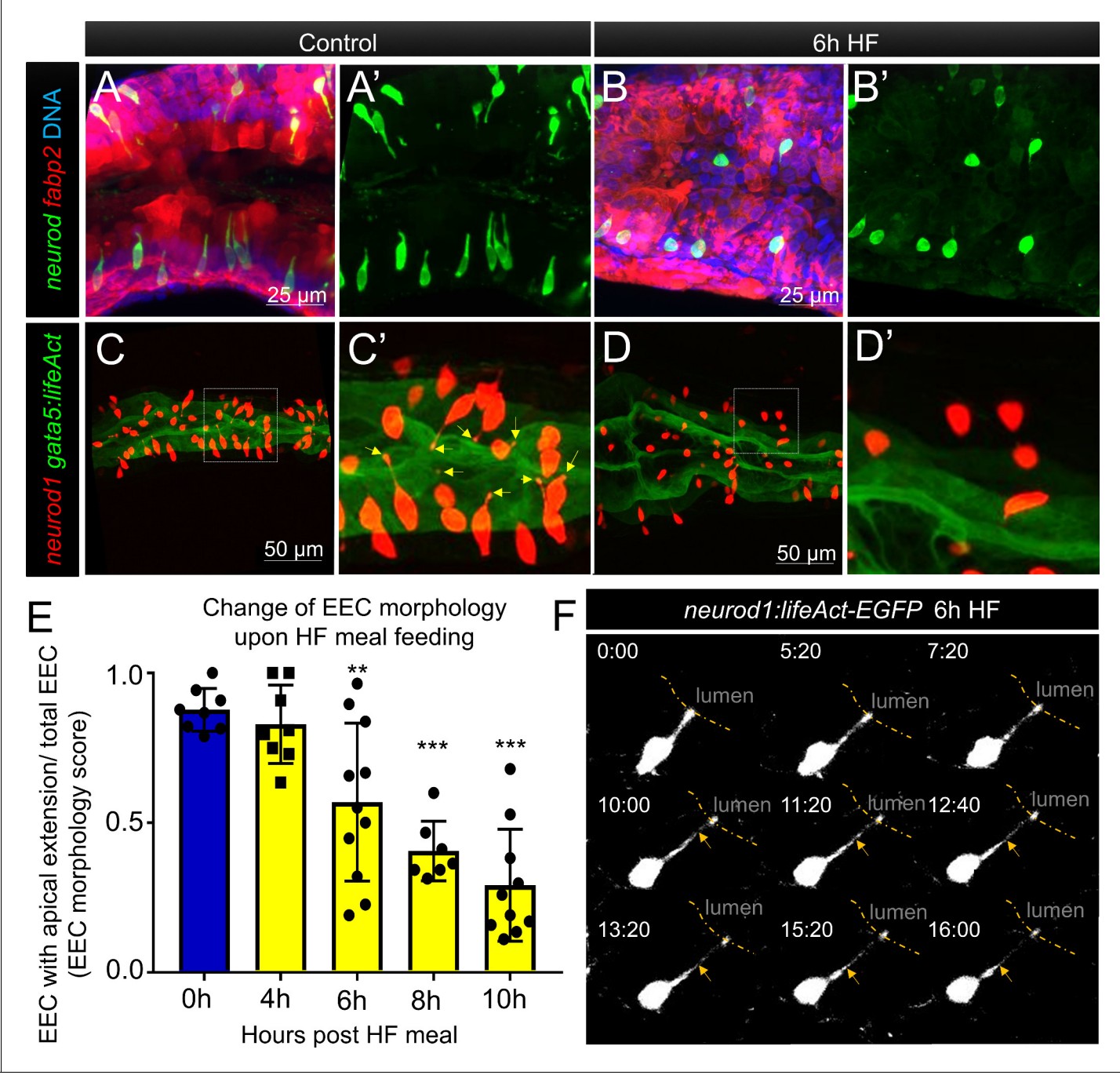

**Figure 4.** Enteroendocrine cells lose their apical extensions following high fat feeding. (**A–B**) Confocal projection of 6 dpf zebrafish intestine in control (**A**) and 6 hr high fat (HF) fed larvae (**B**). Enteroendocrine cells (EECs) are marked by *TgBAC(neurod1:EGFP)* and enterocytes are marked by *Tg(fabp2: DsRed)*. (**A'–B'**) Subpanel images of *neurod1*+EECs in control larvae (**A'**) and 6 hr after high fat feeding (**B'**). (**C–D**) Confocal projection of 6 dpf zebrafish intestine in control (**C**) and 6 hr high fat fed larvae (**D**).The enteroendocrine cells are marked by *Tg(neurod1:RFP)* and the apical region of intestine cells are marked by *Tg(gata5:lifeAct-EGFP)*. (**C'–D'**) Zoom view of EECs in control (**C'**) and HF fed larvae (**D'**). Note that in control intestine, the EECs have extensions that touch the apical lumen (yellow arrow in C'). Such apical extensions in EECs are lost following high fat meal feeding (**D, D'**). (**E**) Quantification of EEC morphology in control and 4–10 hr HF fed zebrafish larvae in *Tg(gata5:lifeAct-EGFP);Tg(neurod1:RFP)* double transgenic zebrafish. The EEC morphology score is defined as the ratio of the number of EECs with apical extensions over the number of total EECs. (**F**) Time lapse images showing loss of the EEC apical extension in 6 hr HF fed larvae using *Tg(neurod1:lifeAct-EGFP)*. One-way ANOVA with post-hoc Tukey test was used in E for statistical analysis. **$p<0.01$, ***$p<0.001$.

The online version of this article includes the following figure supplement(s) for figure 4:

**Figure supplement 1.** HF feeding does not alter EEC morphology in the distal intestine and HF feeding dose not impair *sglt1* expression.

*Figure 4 continued on next page*

Following the photolabeling of EECs, the zebrafish were transferred to fresh egg water and intestines were imaged 20 hr after recovery. Almost all EECs in the recovered zebrafish were labeled with red Kaede (*Figure 5J–K*) indicating that HF feeding did not induce EEC apoptosis. In summary, our data suggest that EECs' morphological and functional adaptations in response to HF feeding are transient and reversible. We operationally define this novel EEC morphological and functional post-prandial adaption to HF feeding as 'EEC silencing'.

## Activation of ER stress following high fat feeding leads to EEC silencing

We next sought to identify the mechanisms underlying HF feeding-induced EEC silencing. Quantitative RT-PCR assays in dissected zebrafish digestive tracts 6 hr after HF feeding revealed broad increases in expression of transcripts encoding EEC hormones (*Figure 6A*). The largest increases were *pyyb* and *ccka* (*Figure 6A*), both of which are expressed by EECs in the proximal zebrafish intestine (*Figure 1*) and are important for the response to dietary lipid. However, HF feeding did not significantly alter expression of EEC specific transcription factors (*neurod1*, *pax6b*, *isl1*), nor the total number of EECs per animal (*Figure 6A and C*). We next assessed how soon after HF feeding EEC hormones were induced. We found that HF feeding led to gradual increases in *ccka* and *pyyb* transcript levelss which plateaued 6 hr after HF feeding (*Figure 6—figure supplement 1E*). Despite these increases in transcript level, PYY immunofluorescence revealed that reduced fluorescence intensity at 5 hr and 8 hr post HF fed zebrafish compared to control zebrafish (*Figure 6—figure supplement 1A–D and F*). This decreased PYY protein content in EECs may be due to depletion of protein contents after HF feeding-induced secretion of hormone or reduced protein translation (*Moran-Ramos et al., 2012*). We speculated that HF feeding challenges existing EECs to increase hormone secretion and synthesis which might place an elevated stress on the endoplasmic reticulum (ER), the organelle where hormone synthesis takes place. Induction of ER stress is known to activate ER membrane sensors Atf6, Perk and Ire1 and a series of downstream cell signaling responses as a negative feedback to block protein translation and reduce ER burden (*Hetz, 2012*; *Xu et al., 2005*). The activated ER stress sensor Ire1 then splices mRNA encoding the transcription factor Xbp1, which in turn induces expression of target genes involved in the stress response and protein degradation, folding and processing (*Yoshida et al., 2001*). Using quantitative RT-PCR analysis in dissected zebrafish digestive tracts, we found that HF feeding increased expression of UPR genes including chaperone proteins Gpr94 and Bip as early as 2 hr after HF feeding (*Figure 6B*, *Figure 6—figure supplement 1E*). To investigate whether ER stress is activated in EECs, we took advantage of a transgenic zebrafish line *Tg(ef1α:xbp1δ-gfp)* that permits visualization of ER stress activation by expressing GFP only in cells undergoing *xbp1* splicing (*Li et al., 2015*). We crossed *Tg(ef1α:xbp1δ-gfp)* with *Tg(neurod1:RFP)* zebrafish and found that zebrafish larvae fed a HF meal, but not control larvae, displayed a significant induction of GFP in *neurod1*+ EECs (*Figure 6L–N*; *Videos 7* and *8*). Next, we tested if activation of ER stress in EECs is required for EEC silencing. Whereas HF feeding normally reduces the EEC morphology score, this did not occur in zebrafish treated with tauroursodeoxycholic acid (TUDCA), a known ER stress inhibitor (*Uppala et al., 2017*; *Vang et al., 2014*) (*Figure 6O–Q*).

To further test the hypothesis that ER stress activation can lead to EEC silencing, we tested if induction of ER stress is sufficient to cause EEC silencing independent of HF feeding. We treated 6 dpf *Tg(neurod1:Gcamp6f)* zebrafish larvae with thapsigargin, a chemical compound commonly used to induce ER stress by interrupting ER calcium storage and protein folding (*Samali et al., 2010*), and then performed the EEC response assay. Thaspisgargin treatment did not alter the basal EEC calcium level in the proximal intestine (*Figure 6—figure supplement 2A–C*). Thapsigargin treatment, however, reduced the EEC calcium response in that region to both glucose and palmitate (*Figure 6D–I*, *Figure 6—figure supplement 2D–E*) and decreased the EEC morphology score, both

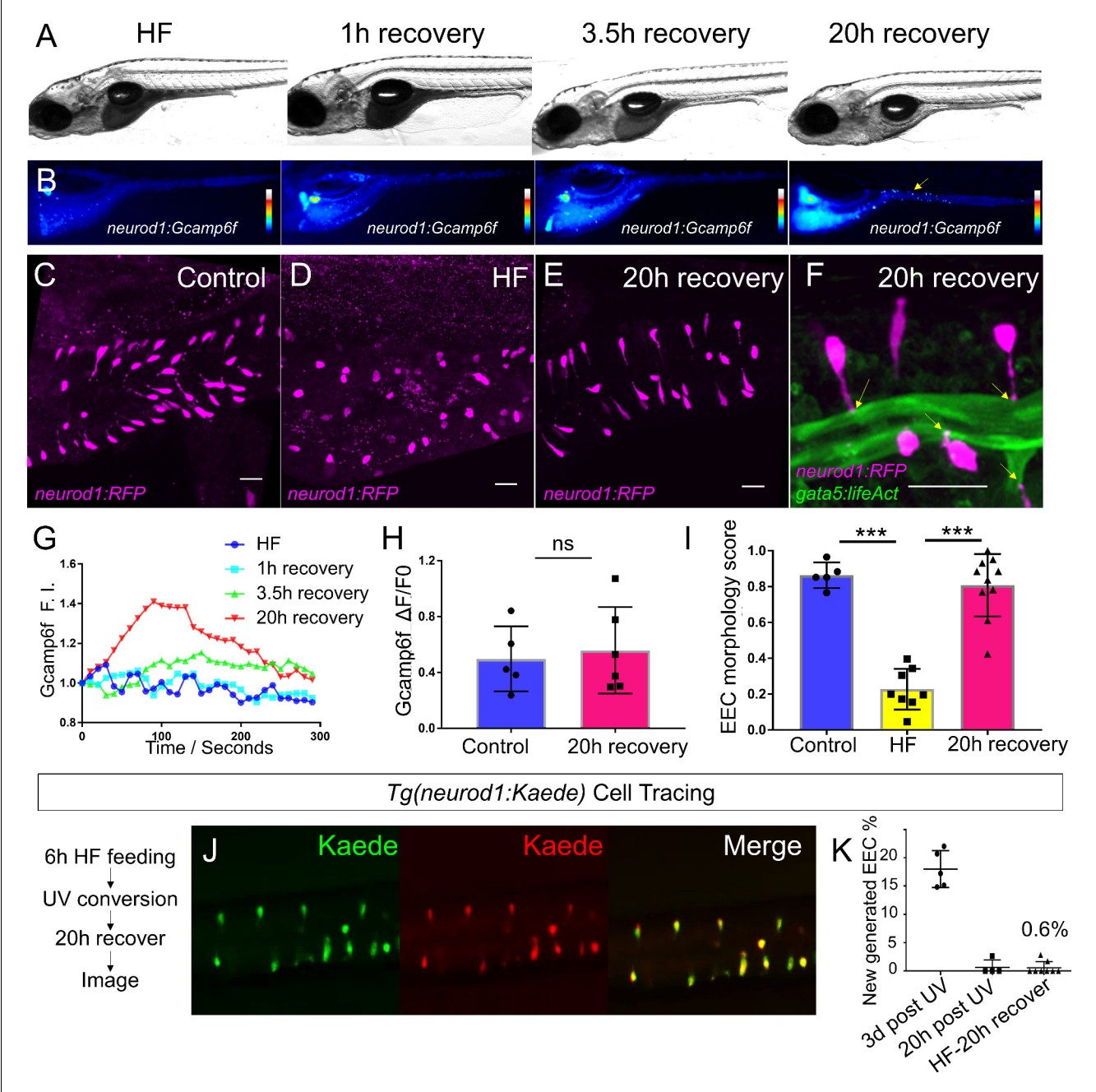

**Figure 5.** High fat feeding induced EEC silencing is reversible. (**A**) Representative image of zebrafish after 6 hr of high-fat (HF) feeding, or HF feeding followed by 1, 3.5, and 20 hr of recovery in fresh egg water. (**B**) EEC palmitate-induced calcium response using *Tg(neurod1:Gcamp6f)* transgenic zebrafish after 6 hr of HF feeding, or HF feeding followed by 1, 3.5, and 20 hr of recovery. (**C–E**) Confocal projection of representative EECs (magenta) in *Tg(neurod1:RFP)* zebrafish under control conditions or 8 hr of HF feeding and HF fed zebrafish following 20 hr of recovery. (**F**) Confocal projection of representative EECs of *Tg(neurod1:RFP); Tg(gata5:lifeAct-EGFP)* in HF fed zebrafish following 20 hr of recovery. Yellow arrows indicate EECs' apical extensions. (**G**) Change of Gcamp6f relative fluoresence intensity in response to palmitate stimulation in HF fed, and HF fed zebrafish following 1, 3.5, and 20 hr of recovery. (**H**) Quantification of EEC palmitate response in control and HF fed zebrafish following 20 hr of recovery. (**I**) Quantification of EEC morphology in control, HF fed and HF fed zebrafish following 20 hr of recovery. (**J**) Representative image of HF fed *Tg(neurod1:Kaede)* zebrafish following 20 hr recovery. *Kaede+* EECs are photoconverted at 6 hr post HF feeding before and after recovery. (**K**) Quantification of the percentage of newly generated EECs (green Kaede only) in 3d post UV photoconversion, 20 hr post UV photoconversion and in HF fed zebrafish photoconverted before 20 hr recovery. Student t-test was used in H and one-way ANOVA with post-hoc Tukey test was used in I for statisitical analysis. ***p<0.001, ns p>0.05, not signficantly different.

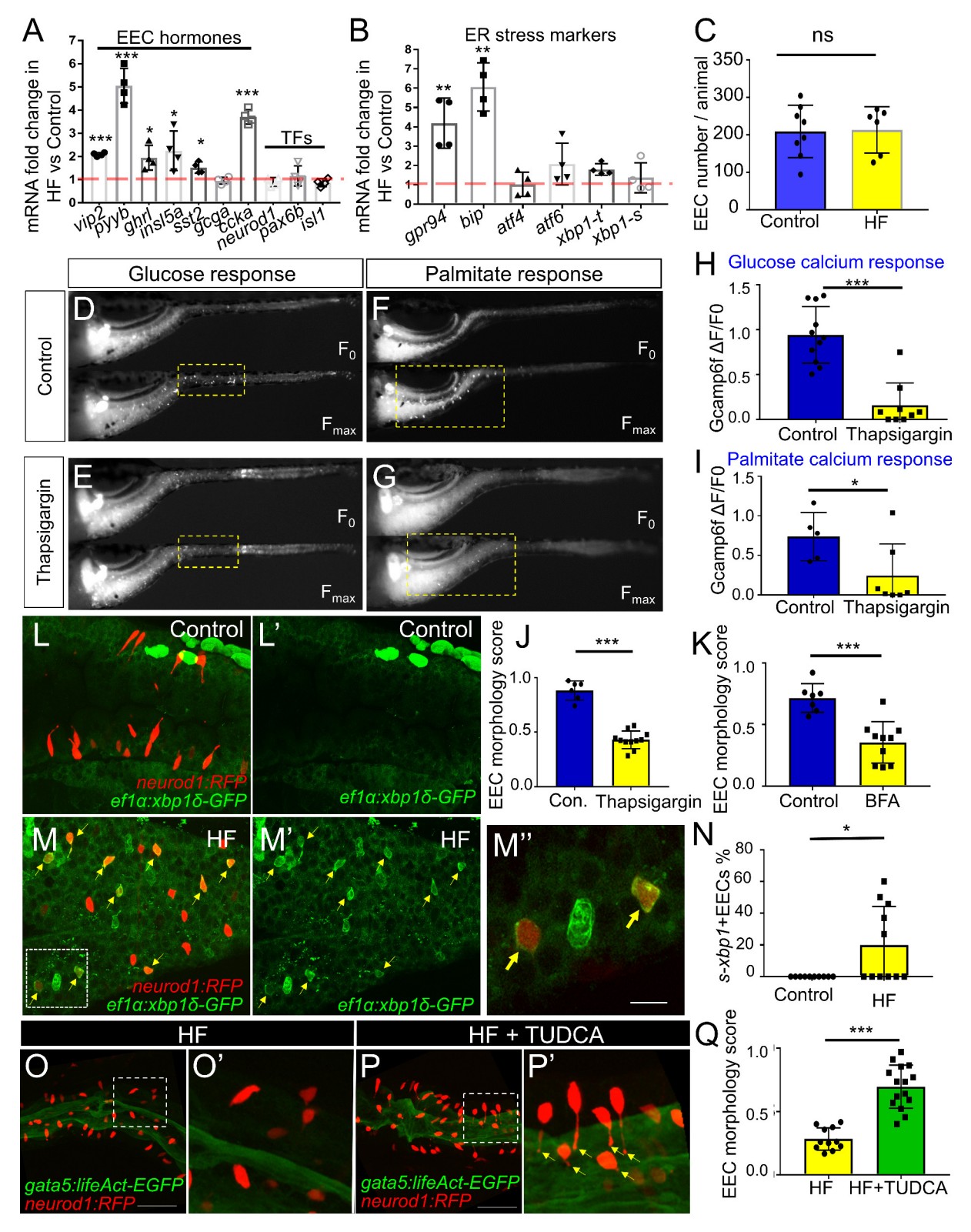

**Figure 6.** Activation of ER stress following high fat feeding leads to EEC silencing. (**A–B**) Quantitative real-time PCR measurement of relative mRNA levels from dissected digestive tracts in control and 6 hr high fat (HF) meal larvae at 6dpf (n = 4 biological replicate pools of 20 fish per condition). The plot indicates the fold increase of relative mRNA levels of indicated genes. (**C**) Quantification of total EEC number in control (n = 8) and 6 hr HF fed larvae (n = 6). (**D–G**) Representative images of the EEC calcium response to glucose or palmitate stimulation in control (**D, F**) and 2 hr thapsigargin (ER

*Figure 6 continued on next page*

*Figure 6 continued*

stress inducer, 1 µM) treated larvae (E, G). (H, I) Quantification of the EEC calcium response toward glucose (H) and palmitate (I) in control and 2 hr thapsigargin (1 µM) treated larvae. (J–K) Quantification of EEC morphology score in control and 10 hr thapsigargin (0.75 µM) or brefeldin A (BFA, 9 µM) treated larvae *Tg(gata5:lifeAct-EGFP);Tg(neurod1:RFP)* double transgenic line. (L–M) Confocal projections of control (J) and 6 hr HF fed (K) zebrafish intestines. The EECs are marked by *Tg(neurod1:RFP)*, the activation of ER stress is marked by *Tg(ef1α:xbp1δ-GFP)* and DNA is stained with Hoechst 33342 (blue). (L'–M') Subpanel images showing the activation of ER stress in control (L') and 6 hr HF fed (M') zebrafish intestines. (M'') Zoom in view of *s-xbp1+* EECs that displayed typical closed morphology in HF fed zebrafish intestine. Yellow arrows in M, M' and M'' indicate EECs with *xbp1* activation. (N) Quantification of *s-xbp1+* EECs (%) in control and 6 hr HF fed zebrafish larvae. (O–P) Confocal projection of zebrafish intestine in 10 hours HF fed (O) and 10 hr HF fed treated animals receiving 0.5 mM TUDCA (P). EECs are marked with *Tg(neurod1:RFP)* and the apical region of the intestine is marked with *Tg(gata5:lifeAct-EGFP)*. (O'–P') Zoom view of EECs in indicated conditions. Yellow arrows in P' indicate EECs' apical extensions. (Q) Quantification of the EEC morphology score in zebrafish larvae following 10 hr of HF feeding and 10 hr of HF feeding with 0.5 mM TUDCA. Student t-test was performed for statistical analysis. *p<0.05, **p<0.01, ***p<0.001.

The online version of this article includes the following figure supplement(s) for figure 6:

**Figure supplement 1.** EEC temporal response to HF feeding.
**Figure supplement 2.** Treatment of Thapsigargin inhibited EEC response to nutrient stimulation.

key phenomena associated with EEC silencing (*Figure 6J*). To confirm this result, we tested a second ER stress inducer brefeldin A (BFA), which inhibits anterograde ER export to Golgi and blocks protein secretion (*Donaldson et al., 1992*; *Klausner et al., 1992*). Similar to thapsigargin, treatment with BFA significantly decreased the EEC morphology score (*Figure 6K*). These results support a working model wherein increased hormone synthesis and secretion following HF feeding induce ER stress in EECs which leads to EEC silencing.

## Blocking fat digestion and absorption inhibits EEC silencing following high fat feeding

We next sought to explore the physiological mechanisms within the gut lumen that may lead to EEC silencing after HF feeding. We reasoned that induction of ER stress in EECs after a HF meal is likely caused by over-stimulation with fatty acids and other nutrients derived from the meal. Fatty acids are liberated from dietary triglycerides in the gut lumen through the activity of lipases, so we predicted that lipase inhibition would block EEC silencing normally induced by HF feeding. We therefore treated zebrafish larvae with orlistat, a broad-spectrum lipase inhibitor commonly used to treat obesity (*Ballinger, 2000*; *Hill et al., 1999*). We found that treatment of *Tg(neurod1:Gcamp6f)* zebrafish with orlistat during HF feeding significantly increased the ability of EECs to subsequently respond to glucose and palmitate (*Figure 7A–F*). Next, we investigated the effect of orlistat on EEC morphology during HF feeding in *Tg(gata5:lifeAct-EGFP); Tg(neurod1:RFP)* zebrafish. We found that following 10 hr of HF feeding, EECs in control animals had switched from an open-type to a closed-type morphology and significantly reduced the EEC morphology score (*Figure 7G and N*). By contrast, treatment with orlistat prevented HF induced EEC morphological changes (*Figure 7H and N*), suggesting lipase activity is required for EEC silencing.

To investigate further how orlistat treatment inhibits EEC silencing, we analyzed its effect on ER stress in EECs following HF feeding using *Tg(ef1α:xbp1δ-gfp)* zebrafish. We found that orlistat treatment significantly reduced the percentage of EECs that are *ef1α:xbp1δ-gfp+* following HF feeding (*Figure 7I,J and O*). We next sought to test if additional pathways are activated in EECs by HF feeding, and if those EEC responses are dependent on lipase activity or ER stress. Induction of ER stress can lead to activation of the transcription factor NF-κB through release of calcium from the ER, elevated reactive oxygen intermediates or direct Ire1 activity (*Kim et al., 2015*; *Pahl and Baeuerle, 1997*). After crossing a transgenic reporter of NF-κB activity *Tg(NFkB:EGFP)* (*Kanther et al., 2011*) with *Tg(neurod1:RFP)*, we found that HF feeding significantly increased the number of NF-κB+ EECs (*Figure 7K and P*), but that this effect could be significantly reduced by treatment with orlistat or the ER stress inhibitor TUDCA (*Figure 7L,M and P*). These results indicate that EEC silencing and associated signaling events that follow ingestion of a HF meal require lipase activity.

Lipases act on dietary triglycerides to liberate fatty acids and monoacylglycerols that are then available for stimulation of EECs (*Hara et al., 2011*; *Lauffer et al., 2009*). To test if free fatty acids are sufficient to induce EEC silencing, we treated 6 dpf zebrafish larvae with palmitate, a major fatty acid component in our HF meal (*Poureslami et al., 2012*). Treatment with palmitate for 6 hr

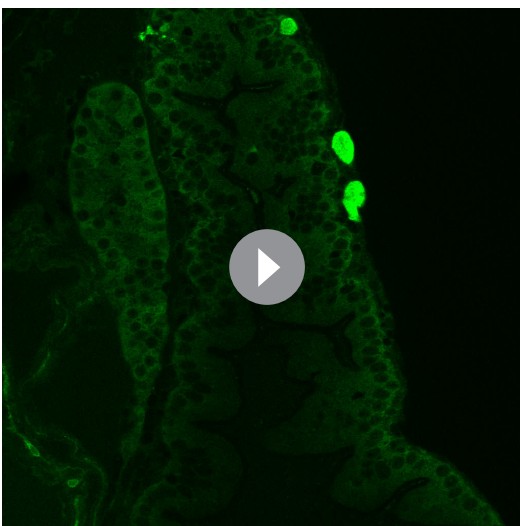

**Video 7.** Confocal Z stack image of *Tg(ef1α:xbp1δ-GFP)* control zebrafish intestine at 6 dpf.
https://elifesciences.org/articles/48479#video7

significantly reduced the ability of EECs to respond to subsequent palmitate stimulation, but did not influence the EEC morphology score, nor the EEC response toward subsequent glucose stimulation (*Figure 7—figure supplement 1A–E*). Similarly, using *Tg(ef1α:xbp1δ-gfp)* and real-time PCR to examine relative *bip* and *grp94* expression, we found that palmitate treatment did not induce significant ER stress activation like HF feeding (*Figure 7—figure supplement 1F–H*). These results suggest that the fatty acid palmitate is sufficient to induce only a portion of the EEC silencing phenotype induced by a complex HF meal.

## High fat feeding induces EEC silencing in a microbiota dependent manner

Using the same HF feeding model in zebrafish, we previously showed that the gut microbiota promote intestinal absorption and metabolism of dietary fatty acids (*Semova et al., 2012*), and similar roles for microbiota have been established recently in mouse (*Martinez-Guryn et al., 2018*). We therefore predicted that the microbiota may also regulate EEC silencing after HF feeding. Using our established methods (*Pham et al., 2008*), we raised *Tg(gata5:lifeAct-EGFP); Tg(neurod1:RFP)* zebrafish larvae to 6 dpf in the absence of any microbes (germ free or GF) or colonized at 3 dpf with a complex zebrafish microbiota (ex-GF conventionalized or CV). In the absence of HF feeding, we observed no significant differences between GF and CV zebrafish in their EEC morphology score or EEC response to palmitate (*Figure 8C,D,G and I*). We then performed HF feeding in these 6 dpf GF and CV zebrafish larvae. In contrast to CV HF-fed zebrafish larvae, EECs in GF zebrafish did not show a change in morphology after HF feeding

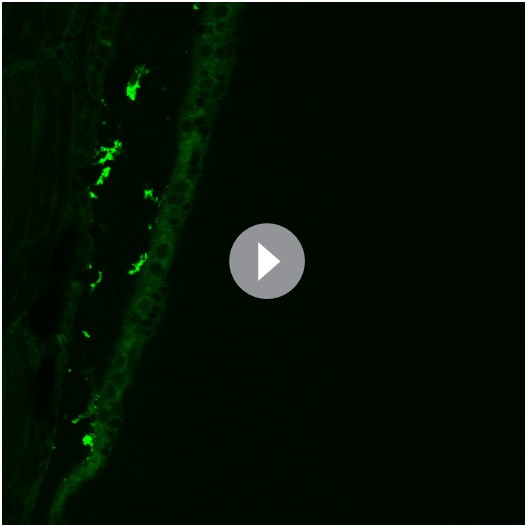

**Video 8.** Confocal Z stack image of *Tg(ef1α:xbp1δ-GFP)* 6 hr high fat fed zebrafish intestine at 6 dpf. Note the induction of *s-xbp1+* cells in the intestinal epithelium.
https://elifesciences.org/articles/48479#video8

(*Figure 8A,B and I*) and exhibited significantly greater responses to palmitate stimulation (*Figure 8E,F and H*). In accord, the ability of HF feeding to induce reporters of ER stress and NF-κB activation was significantly reduced in GF compared to CV zebrafish (*Figure 8J and K*). These results indicate that colonization by microbiota mediates EEC silencing in HF fed zebrafish. EECs are known to express Toll-like receptors (TLRs) (*Kanwal et al., 2014*; *Palti, 2011*), which sense diverse microbe-associated molecular patterns and signal through the downstream adaptor protein Myd88 leading to activation of NF-κB and other pathways (*Kawasaki and Kawai, 2014*). To test if EEC silencing requires TLR signaling, we evaluated *myd88* mutant zebrafish (*Burns et al., 2017*). We found that EECs' response to palmitate after HF feeding was equivalent to that of wild type fish (*Figure 8—figure supplement 1*), suggesting microbiota promote EEC silencing in a Myd88-independent manner.

HF diets are known to significantly alter gut microbiota composition in humans, mice and zebrafish (*David et al., 2014*;

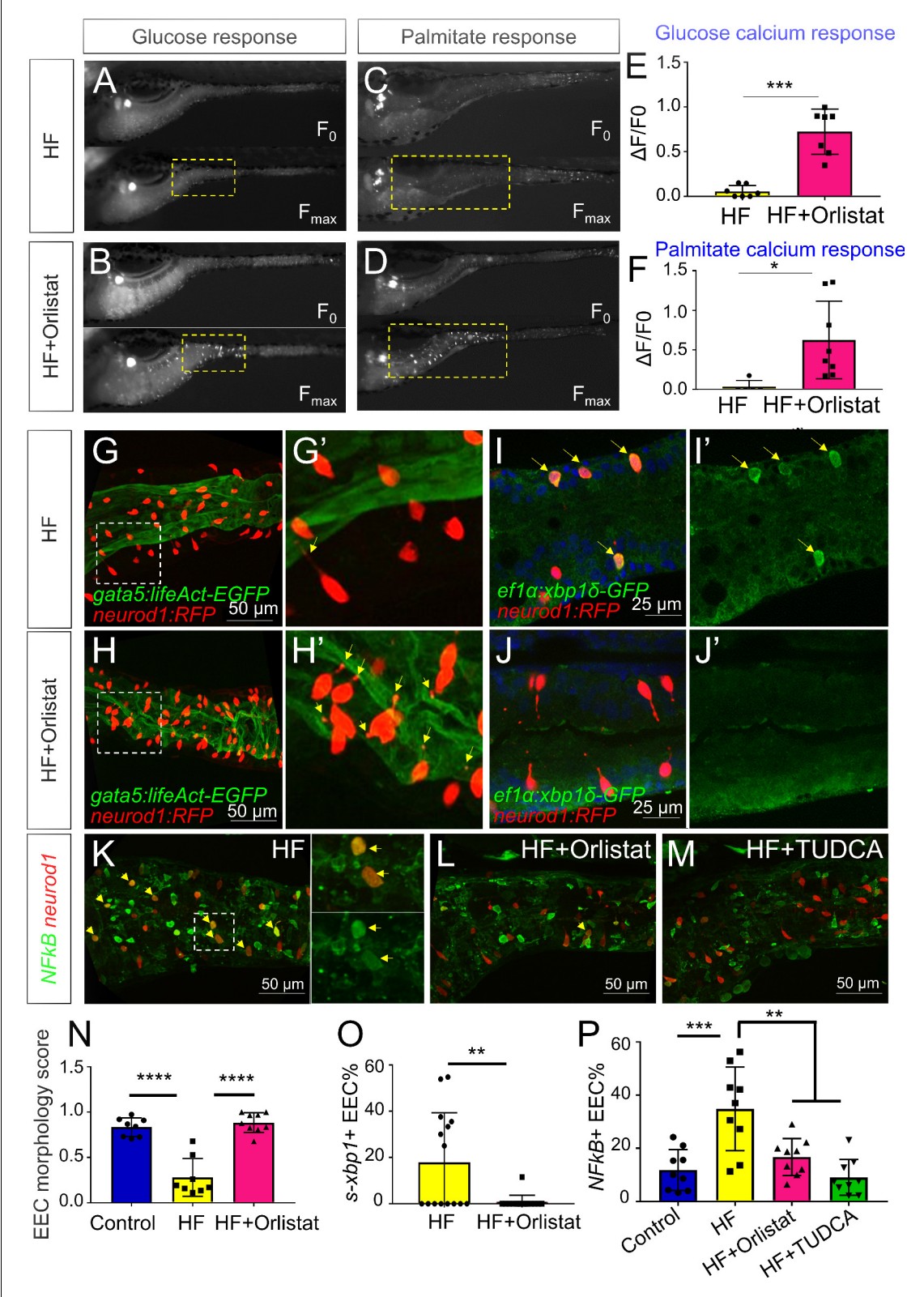

**Figure 7.** Orlistat treatment inhibited high fat feeding induced EEC silencing. (A–D) Representative image of the EEC calcium response to glucose (A, B) and palmitate (C, D) stimulation in 6 hr high fat (HF) fed and 6 hr HF fed with 0.5 mM orlistat treated *Tg(neurod1:Gcamp6f)* zebrafish larvae. (E, F) Quantification of the EEC calcium response to glucose and palmitate stimulation in 6 hr HF fed and 6 hr HF fed with 0.5 mM orlistat treated zebrafish larvae. (G–H) Confocal projection of *Tg(neurod1:RFP); Tg(gata5:lifeAct-EGFP)* zebrafish intestine in 10 hr HF fed larvae (G) and 10 hr HF fed with 0.1 mM

*Figure 7 continued on next page*

*Figure 7 continued*

orlistat treated larvae (H). (G'–H') Zoom view of EECs in indicated conditions. The yellow arrows in G' and H' indicate the EECs' apical extensions. (I–J) Confocal images of *Tg(neurod1:RFP); Tg(ef1α:xbp1δ-GFP)* zebrafish intestine in 6 hr HF fed larvae (I) and 6 hr HF fed with 0.5 mM orlistat treated larvae (J). (I'–J') Zoom view of EECs in indicated conditions. Yellow arrows in I' indicate the EECs with activated *xbp1* in HF fed condition. (K–M) Confocal images of *Tg(neurod1:RFP); Tg(NFKB:EGFP)* zebrafish intestine in 10 hr HF fed larvae (K), 10 hr HF fed larvae treated with 0.1 mM orlistat (L) and 10 hr HF fed larvae treated with 0.5 mM TUDCA (M). Yellow arrows indicate *neurod1:RFP*+ EECs co-labeled with the *NFKB* reporter. (N) Quantification of the EEC morphology score in control, 10 hr HF fed and 10 hr HF fed with 0.1 mM orlistat treated larvae represented in G and H. (O) Quantification of *s-xbp1*+ EEC (%) in 6 hr HF fed larvae and 6 hr HF fed larvae treated with 0.5 mM orlistat represented in J and K. (P) Quantification of NF-κB+ EECs in control, 10 hr HF fed, 10 hr HF fed with 0.1 mM Orlistat and 10 hr HF fed larvae treated with 0.5 mM TUDCA treated zebrafish larvae represented in K-M. Student t-test was performed in E, F, O and one-way ANOVA with post-hoc Tukey test was used in N, P for statistical analysis. *p<0.05, **p<0.01, ***p<0.001, ****p<0.0001.

The online version of this article includes the following figure supplement(s) for figure 7:

**Figure supplement 1.** The effect of palmitate feeding on EEC morphology and function.

---

*Hildebrandt et al., 2009*; *Wong et al., 2015*). We therefore hypothesized that HF feeding might alter the composition of the microbiota, which in turn might promote EEC silencing. To test this possibility, we first analyzed the effects of HF feeding on intestinal microbiota density through colony forming unit (CFU) analysis in dissected intestines from CV zebrafish larvae. Strikingly, we found that intestinal microbiota abundance had increased ~20 fold following 6 hr of HF feeding (*Figure 9A*). To determine if this increase in bacterial density was accompanied by alterations in bacterial community structure, we performed 16S rRNA gene sequencing. Since diet manipulations can alter microbiota composition in the zebrafish gut as well as their housing water media (*Wong et al., 2015*), we analyzed samples from dissected intestines of zebrafish larvae in control and HF fed groups as well as their respective housing media (*Figure 9B*). Analysis of bacterial community structure using the Weighted Unifrac method (*Caporaso et al., 2010*) revealed, as expected, relatively large differences between gut and media samples (PERMANOVA p<0.02 control gut vs. control media, p<0.005 HF gut vs HF media) (*Figure 9C*). The addition of HF feeding had a relatively smaller but consistent effect on overall bacterial community structure in both gut and media (PERMANOVA p=0.2 control gut vs HF gut, p=0.094 control media vs HF media) (*Figure 9C*). HF feeding caused a small reduction in within-sample diversity among media microbiotas as measured by Faith's Phylogenetic Diversity (Kruskal-Wallis p=0.049), but no significant effects on gut microbiotas (p=0.29) (*Faith and Baker, 2007*). Taxonomic analysis of zebrafish gut and media samples revealed several bacterial taxa significantly affected by HF feeding (*Supplementary file 2*). Members of class Betaproteobacteria dominated the control media, but HF feeding markedly decreased their relative abundance (LDA effect size 5.45, p=0.049). Conversely, HF feeding increased the relative abundance of members of class Gammaproteobacteria (LDA effect size 5.49, p=0.049; *Figure 9D*) such as genera *Acinetobacter* (LDA effect size 5.13, p=0.049), *Pseudomonas* (LDA effect size 5.02, p=0.049) and *Aeromonas* (LDA effect size 4.78, p=0.049; *Figure 9F*; *Supplementary file 2 and 3*). HF feeding also increased the relative abundance in media of class Cytophagia from phylum Bacteroidetes (LDA effect size 4.66, p=0.049; *Figure 9D*) due to increases in the genus *Flectobacillus* (LDA effect size 4.76, p=0.049; *Figure 9E*; *Supplementary file 2 and 3*). The increased relative abundances of *Aeromonas* sp. and *Pseudomonas* sp. in HF fed medias was not recapitulated in the gut microbiotas (*Figure 9F*; *Supplementary file 2*). However, similar to the media, HF feeding significantly increased abundance of class Cytophagia (LDA effect size 4.01, p=0.018; *Figure 9D*) due to enrichment of *Flectobacillus* (LDA effect size 4.01, p=0.004; *Figure 9F*). Additionally, HF feeding resulted in a 100-fold increase the relative abundance of *Acinetobacter* sp. in the gut (average 0.04% in control gut, 4.28% in HF gut; LDA effect size 4.31, p=0.001; *Figure 9G*, *Supplementary file 2 and 4*). These results establish that HF feeding has diverse effects on the bacterial communities in the zebrafish gut and media, and raise the possibility that members of these affected bacterial genera may regulate EEC silencing in response to HF feeding.

We next tested if EEC silencing could be facilitated by representative members of the zebrafish microbiota, including those enriched by HF feeding. We selected a small panel of bacterial strains that were isolated previously from the zebrafish intestine (*Stephens et al., 2016*) and used them to monoassociate separate cohorts of GF *Tg(gata5:lifeAct-EGFP); Tg(neurod1:RFP)* zebrafish at 3 dpf (*Figure 9I*). These bacteria strains were from nine different genera including *Acinetobacter* sp.

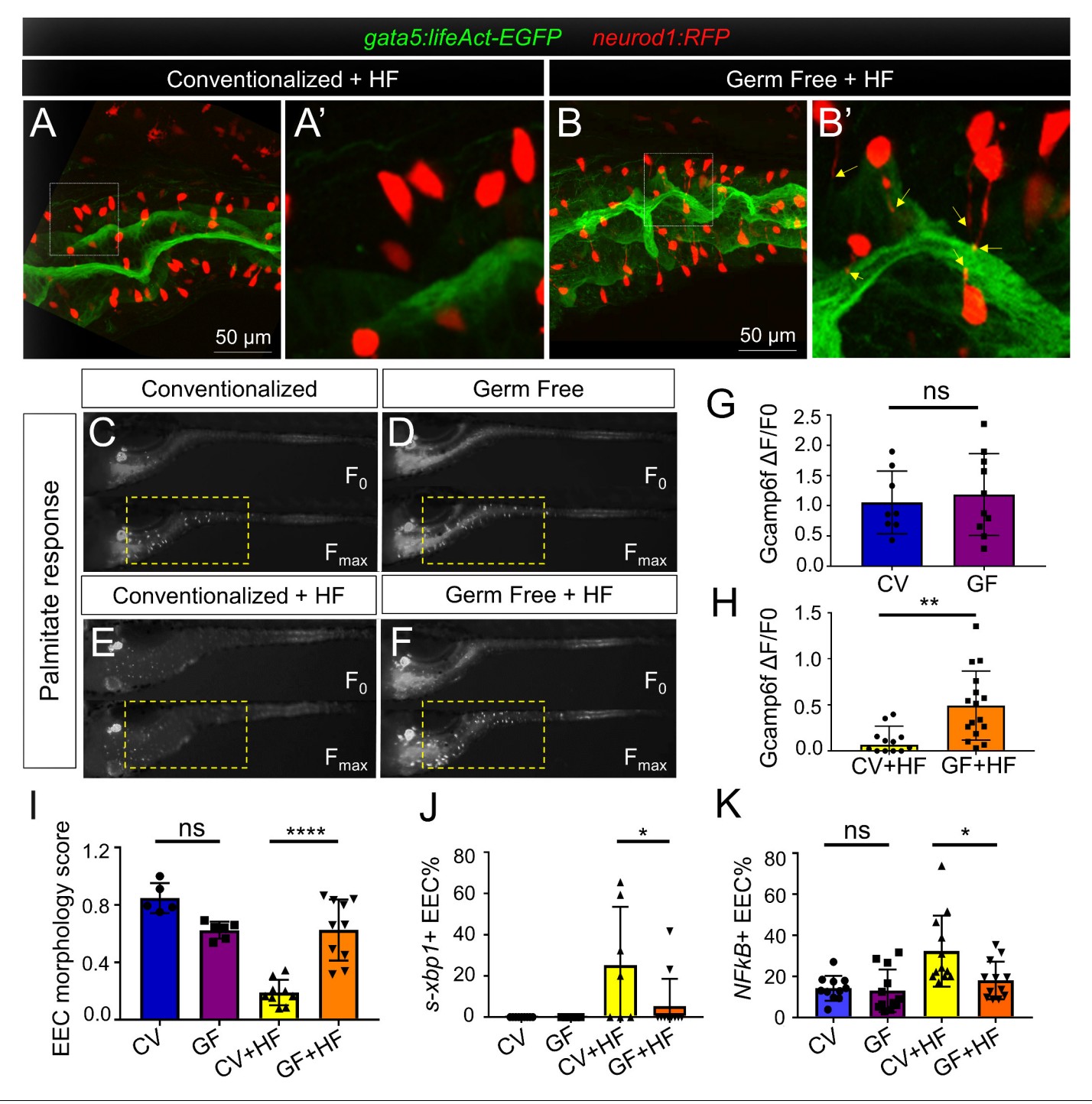

**Figure 8.** High fat feeding induced EEC silencing is microbiota dependent. (**A–B**) Confocal images of 6 dpf zebrafish intestines from conventionalized (CV) and germ free (GF) larvae following 10 hr of high fat (HF) feeding. EECs are marked with *Tg(neurod1:RFP)* and the apical lumen of intestine is marked with *Tg(gata5:lifeAct-EGFP)*. (**A'–B'**) Zoom view of EECs from CV and GF zebrafish following HF feeding. Yellow arrows in B' indicate EEC apical extensions in HF fed GF zebrafish. (**C–F**) Representative images of the EEC calcium response toward palmitate stimulation in CV and GF *Tg(neurod1: Gcamp6f)* larvae with or without 6 hr of HF feeding. (**G–H**) Quantification of the EEC calcium response to palmitate stimulation represented in C-F. (**I**) Quantification of the EEC morphology score in CV and GF zebrafish larvae with or without 10 hr of HF feeding represented in A and B. (**J**) Quantification of *xpb1+* EECs (%) in CV and GF *Tg(neurod1:RFP); Tg(ef1α:xbp1δ-GFP)* zebrafish larvae with or without 6 hr HF feeding. (**K**) Quantification of NF-κB+ EECs (%) in CV and GF *Tg(neurod1:RFP); Tg(NFkB:EGFP)* zebrafish larvae with or without 10 hr HF feeding. Student t-test was used in G,H and one-way ANOVA with post-hoc Tukey test was used in I-K for statistical analysis. *p<0.05, **p<0.01, ****p<0.0001.

*Figure 8 continued on next page*

*Figure 8 continued*

The online version of this article includes the following figure supplement(s) for figure 8:

**Figure supplement 1.** EEC sensitivity to palmitate stimulation is not altered in *myd88* mutant zebrafish.

ZOR0008. We did not observe significant differences in colonization efficiency among these bacteria strains that were inoculated into GF zebrafish (*Figure 9—figure supplement 1*). At 6 dpf, we performed HF feeding and examined the EEC morphology score. Strikingly, only *Acinetobacter* sp. ZOR0008 was sufficient to significantly reduce the EEC morphology score upon HF feeding (*Figure 9J*) similar to conventionalized animals (*Figure 8A,B and I*). Consistently, we also found that monoassociation with *Acinetobacter* sp. ZOR0008 alone is sufficient to reduce EEC calcium response to palmitate stimulation following HF feeding compared with GF controls (*Figure 9—figure supplement 2*). These results indicate that the effects of microbiota on EEC silencing following HF feeding display strong bacterial species specificity, and suggest *Acinetobacter* bacteria enriched by HF feeding may mediate the effect of microbiota on HF sensing by EECs.

Bacterial colonization can increase the production of reactive oxygen species (ROS) by the intestinal epithelium or associated innate immune cells (*Jones et al., 2012*; *Sommer and Bäckhed, 2015*). On the other hand, gram negative bacteria like *Acinetobacter* can also generate ROS through the citric acid cycle and electron transport (*Ajiboye et al., 2018*). The production of ROS and the resulting lipid peroxidation can trigger cellular stress and increase inflammation (*Schieber and Chandel, 2014*). We therefore investigated whether microbiota dependent EEC silencing following HF feeding is triggered by increased intestinal ROS. We first treated conventional raised zebrafish with known ROS scavengers N-acetylcysteine (NAC) (*Mocelin et al., 2019*) and N(ω)-nitro-L-arginine methyl ester (L-NAME) that can inhibit host reactive nitrogen species production (*Bradley et al., 2010*), however neither of these compounds prevented HF feeding induced EEC silencing (*Figure 9—figure supplement 3A–D and G*). Using a general oxidative stress chemical indicator CM-H2DCFDA whose fluorescence can be induced by ROS mediated oxidation, we also failed to detect significant differences in intestinal fluoresence in the conditions of GF, CV and *Acinetobacter* sp. monoassociated zebrafish with or without HF feeding (*Figure 9—figure supplement 3E–F and H*). Similarly we did not observe significant induction of ROS production in in vitro cultures of *Aeromonas* sp. and *Acinetobacter* sp. in media containing HF meal (*Figure 9—figure supplement 3I*). These results suggest that microbiota dependent EEC silencing following HF feeding is not mediated by ROS signaling.

## Discussion

In this study, we established a new experimental system to directly investigate EEC activity in vivo using a zebrafish reporter of EEC calcium signaling. Combining transgenic, dietary and gnotobiotic manipulations allowed us to uncover a novel EEC adaptation mechanism through which high fat feeding induces rapid change of EEC morphology and reduced nutrient sensitivity. We called this novel adaptation 'EEC silencing'. Our results show that EEC silencing following a high fat meal requires lipase activity, is coupled to ER stress, and is reversible. Furthermore, our data suggest that high fat meal induced EEC silencing requires the presence of microbiota and can be promoted by certain bacterial taxa (e.g., *Acinetobacter* sp.). As discussed below, we propose a working model (*Figure 10*) that nutrient over-stimulation from high fat feeding increases EEC hormone secretion and synthesis burden, overgrowth of the gut bacterial community including enrichment of *Acinetobacter* sp., which collectively activate EEC ER stress response pathways and thereby induce EEC silencing. This study demonstrates the utility of the zebrafish model to study in vivo interactions among diet, gut microbes, and EEC physiology. In the future, the mechanisms underlying EEC silencing could be targeted to manipulate EEC adaptations to diet and microbiota to reduce the incidence and severity of metabolic diseases.

### EEC physiology in zebrafish

These studies provide important new tools for studying EECs in the context of zebrafish intestinal epithelial development and physiology. Similar to mammals, fish EECs are thought to arise from

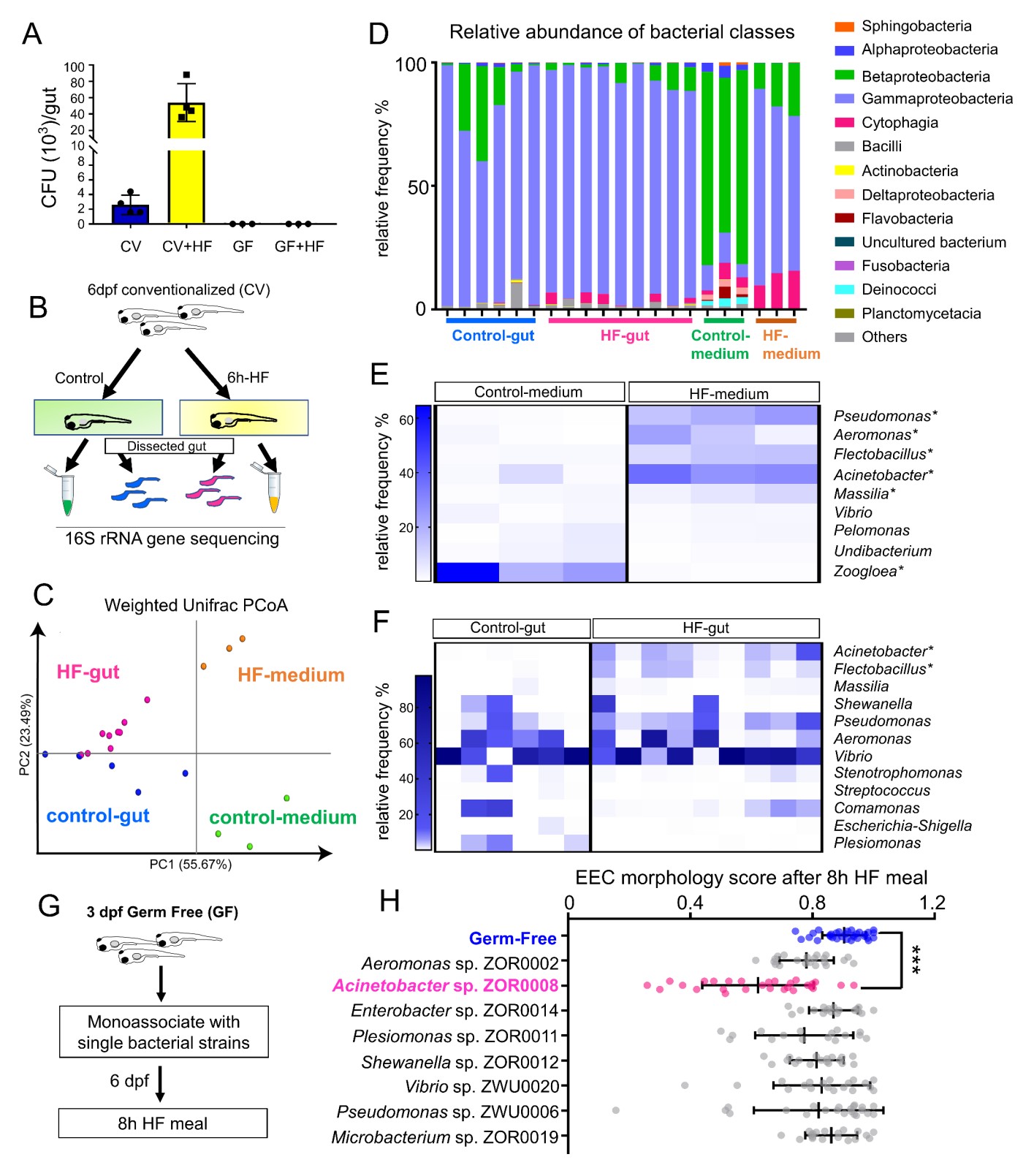

**Figure 9.** High fat feeding modifies microbiota composition. (A) Colony forming unit (CFU) quantification in GF and CV dissected intestines with or without 6 hr of high fat (HF) feeding. (B) Experimental design of 16S rRNA gene sequencing in control larvae dissected gut and medium and 6 hr HF fed larvae dissected gut and medium. (C) Weighted UniFrac principal coordinates analysis (PCoA) of 16S rRNA gene sequences from control and HF fed gut and media samples The % variation explained by principal components (PC) 1 and 2 are shown on their respective axes. (D) Relative abundance

*Figure 9 continued on next page*

*Figure 9 continued*

of bacterial classes in control and HF fed gut and media. (E–F) Change in representative bacterial genera following HF feeding in gut and media. Asterisks indicate taxa with p<0.05 by LEfSe analysis. (G) Schematic of monoassociation screening to investigate the effects of specific bacterial strains on EEC morphology. Three dpf zebrafish larvae were colonized with one of the isolated bacterial strains and EEC morphology was scored after 8 hr high fat meal feeding in 6 dpf GF and monoassociated animals. (H) EEC morphology score of GF and monoassociated zebrafish larvae following 8 hr high fat feeding. Data were pooled from three independent experiments, with each dot representing an individual animal. The EEC morphology score in *Acinetobacter* sp. ZOR0008 monoassociated animals was significantly lower than GF EECs (p<0.001). No consistent significant differences were observed in other monoassociated groups. One way ANOVA followed by Tukey's post-test was used in H for statistical analysis.

The online version of this article includes the following figure supplement(s) for figure 9:

**Figure supplement 1.** Colonization of bacterial strains during monoassociation.
**Figure supplement 2.** *Acinetobacter* sp. ZOR0008 monoassociated zebrafish EECs do not respond to palmitate stimulation after HF feeding.
**Figure supplement 3.** Inhibition of ROS signaling does not prevent HF feeding induced EEC silencing.

intestinal stem cells through a series of signals that govern the differentiation process (*Aghaallaei et al., 2016*). Delta-Notch signaling appears to control the differentiation of stem cells into absorptive and secretory cell lineages in both zebrafish and mammals (*Crosnier et al., 2005*). Activation of Notch signaling can block the differentiation of EECs by inhibiting the expression of key EEC bHLH transcription factors (*Li et al., 2011*). In mammals, the bHLH transcription factor Neurod1 that has been shown to regulate EEC terminal differentiation (*Li et al., 2011*; *Ray and Leiter, 2007*). Our results indicate that Neurod1 is expressed by and important in EEC differentiation in zebrafish as it is in mammals. Moreover, this finding enabled us to use *neurod1* regulatory sequences to label and monitor zebrafish EECs.

The hallmark of EECs is their expression of hormones. In this study, using transgenic reporter lines and immunofluorescence staining approaches to examine a panel of gut hormones in zebrafish EECs, we found that zebrafish EECs express conserved hormones as do mammalian EECs. Interestingly, a subset of EECs express proglucagon peptide which can be processed to hormones glucagon like peptide 1 (GLP-1) and 2 (GLP-2) (*Sandoval and D'Alessio, 2015*). The incretin GLP-1 is released by EECs in response to oral glucose intake and facilitates insulin secretion and reduces blood glucose (*Drucker et al., 2017*). Multiple studies suggest that the expression of Sglt1 is important for EEC glucose sensing (*Gorboulev et al., 2012*; *Reimann et al., 2008*; *Röder et al., 2014*). EECs in Sglt1 knockout mice fail to secrete GLP-1 in response to glucose and galactose (*Gorboulev et al., 2012*). In our studies, we identified similar Sglt1 mediated glucose sensing machinery in zebrafish EECs. This suggests that zebrafish EECs may exhibit conserved roles in regulating glucose metabolism.

Our data also establish that zebrafish EECs develop striking regional specificity in the hormones they express along the intestine (*Figure 1—figure supplement 1*). For example, the CCK and PYY hormones that are important for regulating food digestion and energy homeostasis (*Beglinger and Degen, 2006*; *Liddle, 1997*; *Raybould, 2007*) were only expressed in the proximal intestine. In addition to hormonal regional specificity, we found that the EEC calcium responses to nutrients also display regional specificity. For example, glucose and long chain/medium chain fatty acids only stimulate EECs in the proximal intestine, a region in zebrafish where digestion and absorption of dietary fats primarily occurs (*Carten et al., 2011*). This hormonal and functional regional specificity suggests that distinct developmental and physiological programs govern EEC function along the intestinal tract, and that EECs in the proximal zebrafish intestine may play key roles in monitoring and adapting to dietary nutrients.

## EEC silencing

In this study, we adopted a high fat feeding paradigm that is the most commonly used high fat diet in zebrafish larvae and adults for metabolic and obesity studies (*Maddison and Chen, 2012*; *Minchin et al., 2018*; *Zang et al., 2018*; *Zhou et al., 2015*). This high fat feeding paradigm consisting of 5% chicken egg yolk provides a rich source of dietary lipid (60% lipid by dry weight), and is a common dietary constituent for humans and other animals (*Kuksis, 1992*). We discovered that high fat feeding can induce a series of functional and morphological changes in EECs we refer to as 'EEC silencing'. EEC silencing includes (1) reduced EEC sensitivity to nutrient stimulation (e.g., fatty acids and glucose) and (2) conversion of EEC morphology from an open to a closed type. To our

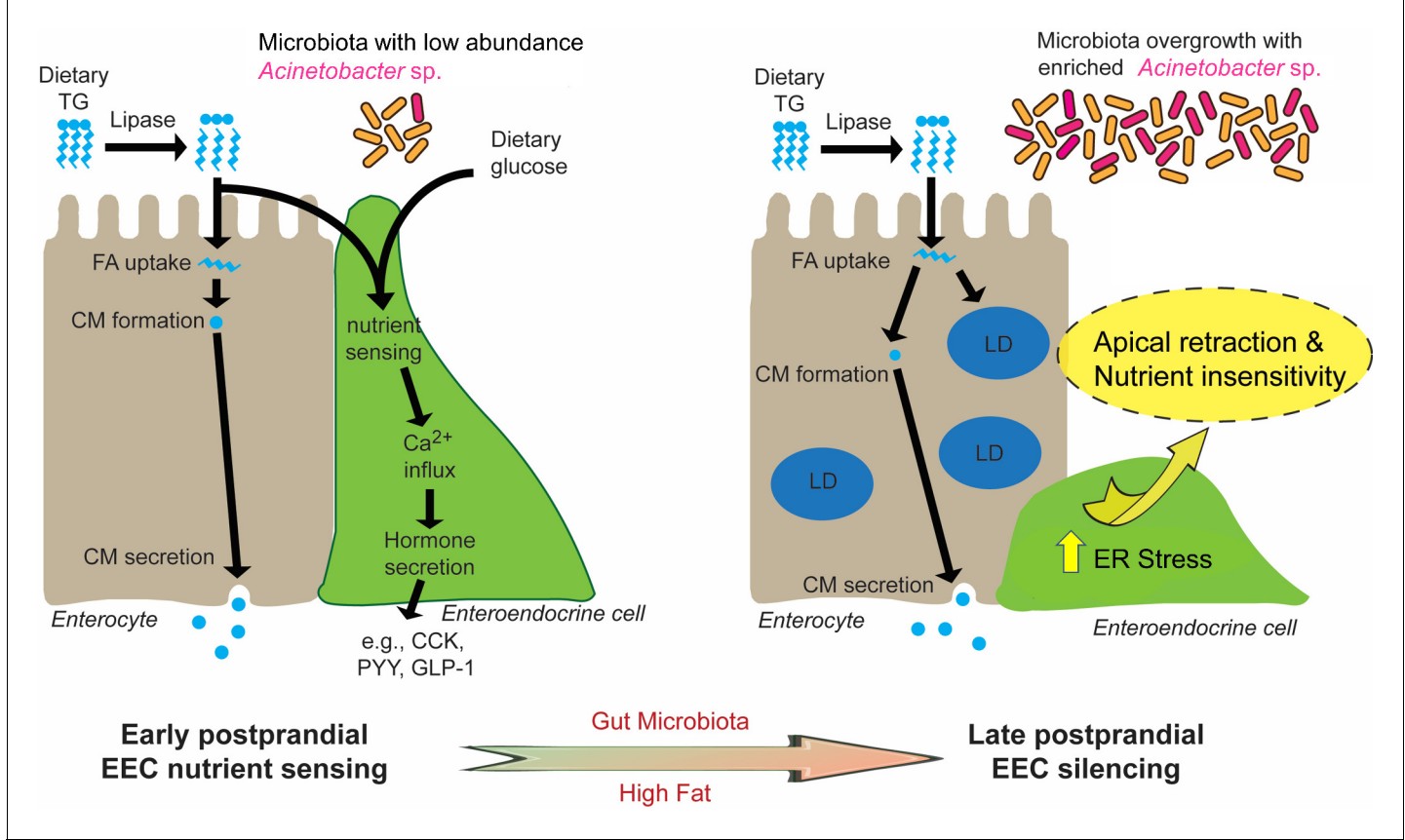

**Figure 10.** Proposed model for microbiota-dependent HF feeding-induced EEC silencing. At early postprandial stages after consumption of a high fat (HF) meal, dietary triglyceride (TG) is hydrolyzed to monoglycerides and free fatty acids (FA) by lipases in the gut lumen. FA are taken up by enterocytes and re-esterified into TG which is packaged into chylomicrons (CM) for basolateral secretion. FA and dietary glucose stimulate EECs, increasing $[Ca^{2+}]_i$ and inducing secretion of hormones like CCK, PYY and GLP-1. During and after HF feeding, FA taken up by enterocytes are stored in cytosolic lipid droplets (LD) in addition to secreted CM. Moving into later postprandial stages, HF feeding and presence of gut microbiota lead to ER stress in EECs. HF feeding also promotes overgrowth of the gut bacterial community including enrichment of *Acinetobacter* sp. Activation of ER stress pathways by these nutritional and microbial stimuli cause EECs to retract their apical processes and reduce their nutrient sensitivity at the late postprandial stage, a process we call 'EEC silencing'.

knowledge, EEC silencing has not been observed in previous studies of EEC in any vertebrate. This underscores the unique power of in vivo imaging in zebrafish to reveal new physiologic and metabolic processes. Our results also demonstrated that EECs' morphological and functional changes in response to HF feeding are reversible and reflect the recovery of pre-existing EECs. This together with other data presented here indicate that EEC silencing is a physiologically relevant postprandial adaptation, rather than acute toxicity in EECs stimulated by high fat feeding. Our evidence further suggests that EEC silencing is a response that EECs display following consumption of a high fat meal only in the presence of specific microbes. The physiologic function of EEC silencing remains unknown. EEC silencing might serve to protect EECs against excessive stress following consumption of a high fat meal. In neurons for example, similar desensitization has been shown to protect nerve cells from excitatory neurotransmitter induced toxicity (*Gainetdinov et al., 2004*; *Quick and Lester, 2002*) and blocking desensitization of excitatory neuronal receptors induces rapid neuronal cell death (*Walker et al., 2009*). High dietary fat can also lead to excessive production of excitatory stimuli like long-chain fatty acids. We speculate that EEC silencing provides an adaptive mechanism for EECs to avoid excessive stimuli and protect against cell stress and death.

The observation that EECs exhibited reduced sensitivity to oral glucose following high fat feeding is interesting and consistent with the finding in mice that high fat feeding reduces intestinal glucose sensing and glucose induced GLP-1 secretion in vivo (*Bauer et al., 2018*). In vitro, small intestinal

cultures from high fat fed mice also exhibit reduced secretory responsiveness to nutrient stimuli including glucose when compared with cultures from control mice (*Richards et al., 2016*) but underlying mechanisms remained unclear. These studies, together with our results, indicate that high fat feeding impairs EEC function. However, how high fat feeding reduces EEC glucose sensitivity is still unclear as we did not detect changes in EEC glucose sensor *sglt1* expression in high fat fed intestine (*Figure 4—figure supplement 1I*). It is possible, however, that high fat feeding affects EECs glucose sensing by altering Sglt1 activity (*Ishikawa et al., 1997*; *Subramanian et al., 2009*; *Wright et al., 1997*). We also speculate that high fat feeding induced EEC morphological changes may contribute to EEC glucose insensitivity. Since Sglt1 is expressed on the brush border at the apical surface of the cell, as EECs switch from an open to closed type morphology they would lose their contact with the gut lumen and exposure to luminal glucose stimuli. It will be interesting to determine if HF feeding induced EEC silencing occurs in mammals, and if it helps explain the ability of HF feeding to impair the incretin effect (*Richards et al., 2016*).

Our observation that EECs can change their morphology from an 'open' to 'closed' state upon high fat feeding was surprising. The majority of EECs in the intestinal tract are open with an apical extension and microvilli facing the intestinal lumen. In contrast, some EECs lie flat on the basement membrane and are 'closed' to the gut lumen (*Gribble and Reimann, 2016*). The presence of open and closed EECs has been observed in both mammals and fish (*Rombout et al., 1978*). Previously, it was believed that the open and closed EECs were two differentiated EEC types that perhaps had different physiological functions (*Gribble and Reimann, 2016*). The open EECs were thought to sense and respond to luminal stimulation while, although less clear, the closed EECs were thought to respond to hormonal and neuronal stimulation from the basolateral side. However, our data reveal that individual EECs can convert reversibly from an open to a closed state. This indicates that EECs possess plasticity to actively prune their apical extensions. The pruning of cellular process can be observed extensively in neurons. Studies from multiple organisms revealed that sensory neurons can eliminate their dendrites and axons during development and in response to injury through active pruning (*Kanamori et al., 2013*; *Nikolaev et al., 2009*; *Sagasti et al., 2005*; *Williams et al., 2006*; *Yu and Schuldiner, 2014*). This process includes focal disruption of the microtubule cytoskeleton, followed by thinning of the disrupted region, severing and fragmentation and retraction in proximal stumps after severing events (*Williams and Truman, 2005*). In our system, the thinning and fragmentation in the EEC apical extension was also observed. It is well known that EECs possess many neuron-like features including neurotransmitters, neurofilaments, and synaptic proteins (*Bohórquez et al., 2015*). Whether EECs adopt the same mechanisms as neurons to prune their cellular processes in response to nutritional and microbial signals is intriguing and requires future study.

Our results reveal important roles for fat digestion in the induction of EEC silencing. Blocking fat digestion and subsequent lipid absorption through orlistat treatment prevented EEC silencing after high fat feeding. EEC function may be directly influenced by the products of lipolysis such as free fatty acids (*Edfalk et al., 2008*; *Hirasawa et al., 2005*; *Katsuma et al., 2005*). However, in our experiments, palmitate treatment was only sufficient to reproduce a portion of the EEC silencing phenotype (i.e. loss of palmitate sensitivity without elevation of ER stress nor change of EEC morphology). These differences in the EEC response to palmitate and a complex high fat meal could have several potential causes. Lipolysis of complex dietary fats yields fatty acid substrates like palmitate that stimulate free fatty acid receptors on EECs. Previous studies demonstrated that repeated or continuous stimulation of G-protein coupled receptors (GPCR) like free fatty acid receptors induces GPCR desensitation through receptor internalization into vesicles, degradation in lysosomes, and decreased receptor mRNA levels (*Rajagopal and Shenoy, 2018*). It is therefore possible that palmitate treatment or high fat meal induces free fatty acid receptor desensitization which prevents EECs' response to further fatty acid stimulation. On the other hand, our data further indicate that high fat feeding but not palmitate treatment induced sustained ER stress in the digestive tract. The ER stress induced by high fat feeding required the presence of gut microbiota, and likely drives other EEC silencing phenotypes including altered EEC morphology or reduced glucose sensitivity. We find that ER stress markers are evident in EECs within 2 hr after high fat feeding, concomitant with increased hormone transcription, whereas EEC silencing is not established until 6 hr. The continuous ER stress which is induced throughout the high fat feeding as early as 2 hr appears to be a key mechanism leading to the later EEC silencing response. The specific molecular components that trigger ER stress in EECs in this model are yet to be identified. We speculate the signal(s) that promote

ER stress in EECs either derives from other nutrients in the intestinal lumen or neighboring cells. In addition to free fatty acid, the digestion of dietary fats in the intestinal lumen increases local concentrations of glycerol, mono-acylglycerol, di-acylglycerol, cholesterol, sphingolipid as well as the complex lipid derivatives from microbial metabolism. These complex lipid species may directly or indirectly act on EECs to induce EEC ER stress and thereby promote EEC silencing. EEC silencing might also be caused by signals from neighboring cells. Within the intestinal epithelium, EECs are surrounded by absorptive enterocytes and these two cell types exhibit complex bi-directional communication (*Hein et al., 2013*; *Hsieh et al., 2009*; *Okawa et al., 2009*; *Shimotoyodome et al., 2009*). Following ingestion of a complex high fat meal, free fatty acids and glycerol liberated from triglyceride digestion are taken up by enterocytes and assembled into lipid droplets and chylomicrons (*Phan, 2001*). The subsequent enlargement of enterocytes from lipid droplet accumulation may exert mechanical pressure on EECs that could force the morphological changes associated with EEC silencing. Besides mechanical pressure, lipoproteins and free fatty acids released from enterocytes may act on EECs basolaterally to alter their function (*Chandra et al., 2013*; *Okawa et al., 2009*; *Shimotoyodome et al., 2009*).

## The effects of diet and microbes on EEC silencing

In this study, we have shown that both diet and microbes play important roles in inducing EEC silencing. Dietary manipulations and changes in gut microbiota have been shown to affect EEC cell number and GI hormone gene expression in mice and zebrafish (*Arora et al., 2018*; *Rawls et al., 2004*; *Richards et al., 2016*; *Troll et al., 2018*). However, it remains unclear from previous studies how environmental factors like diet and microbiota affect EEC function. We found that while the presence of microbiota did not influence EEC nutrient sensing under basal conditions, microbiota played an essential role in mediating EEC silencing as germ free EECs were resistant to high fat diet induced silencing. We speculate that EEC silencing may temporarily attenuate the host's ability to accurately sense ingested nutrients and thereby control energy homeostasis. Our finding that gut microbiota play an essential role in high fat diet induced EEC silencing may provide a new mechanistic inroad for understanding the effects of gut microbiota in diet induced metabolic diseases including obesity and insulin resistance (*Bäckhed et al., 2007*; *Rabot et al., 2010*).

There are several nonexclusive ways by which specific gut microbiota members such as *Acinetobacter* sp. might affect EECs in the setting of a high fat diet. First, microbiota could affect EEC development to increase production of EEC subtypes that are relatively susceptible to diet-induced EEC silencing. In mice, microbiota colonization reduced expression of genes associated with synaptic cycling, ER stress response and cell polarity in GLP-1 secreting EECs (*Arora et al., 2018*). This suggests that EECs in colonized animals may be more prone to diet-induced ER stress and morphological changes including those associated with EEC silencing.

Second, high fat meal conditions induce bacterial overgrowth and alter the selective pressures within the gut microbial community to allow for enrichment and depletion of specific bacterial taxa. Such changes in microbial density and community composition may then acutely affect EEC physiology. Indeed, we found that high fat feeding altered the relative abundance of several bacterial taxa in the zebrafish gut and media, including a 100-fold increase of the *Acinetobacter* genus. Strikingly, a representative *Acinetobacter* sp. was the only strain we identified that was sufficient to mediate high fat induced alterations in EEC morphology. We speculated that bacterial overgrowth may also result in increased presentation of microbe-associated molecular patterns which could then hyperactivate Toll-like receptor (TLR) or other microbe-sensing pathways that could lead to EEC functional changes. However, our data from *myd88* mutant zebrafish suggest that Myd88-dependent microbial sensing pathways are not required for high fat induced EEC silencing. In addition to TLR signaling pathway, our data suggest that microbial or host derived ROS production is not involved in HF feeding induced EEC silencing. As described below, identification of the specific signals produced by *Acinetobacter* sp. and other bacteria that facilitate EEC silencing remain an important research goal.

Third, gut microbiota might affect EEC function by promoting lipid digestion and absorption. As discussed above, our data suggest that fat digestion and absorption is required for EEC silencing. Previous studies in gnotobiotic zebrafish and mice have shown that lipid digestion and absorption is impaired in germ-free animals and enterocytes in germ-free conditions exhibit reduced lipid droplet accumulation (*Martinez-Guryn et al., 2018*; *Semova et al., 2012*). Resistance of germ-free zebrafish to high fat induced EEC silencing might be linked to reduced lipid droplet accumulation in their

enterocytes thereby minimizing increases in mechanical pressure or secondary signaling molecules imposed by enterocytes on their neighboring EECs. In order to understand how microbiota promote EEC silencing, it is important to define the causative microbial species and factors. *Acinetobacter* was the most highly enriched genus in the larval zebrafish intestine following high fat feeding in this study and was also enriched in adult zebrafish gut following a chronic high fat diet (*Wong et al., 2015*). Further, we identified a representative member of this genus that is sufficient to mediate EEC silencing under high fat diet conditions. However, the molecular mechanisms by which *Acinetobacter* spp. evoke this host response remain unknown. Studies suggest that *A. baumannii*, a related oportunitistic pathogen, can signal to host epithelial cells through secreted outer membrane vesicles (OMVs) and activation of downstream inflammatory pathways (*Jha et al., 2017*; *Jin et al., 2011*; *Jun et al., 2013*; *March et al., 2010*). In addition to OMVs, *Acinetobacter* strains are known to secrete phospholipase that can affect host cell membrane stability and interfere with host signaling (*Lee et al., 2017*; *Songer, 1997*). Members of the *Acinetobacter* genus are also known to possess potent oil degrading and lipolytic activities (*Lal and Khanna, 1996*; *Snellman and Colwell, 2004*). Moreover, species from *Acinetobacter* genus have the ability to produce emulsifiers which might enhance lipid digestion (*Navon-Venezia et al., 1995*; *Toren et al., 2001*; *Walzer et al., 2006*). *Acinetobacter* spp. in the human gut are positively associated with plasma triglycerides and total- and LDL-cholesterol (*Graessler et al., 2013*), and *Acinetobacter* spp. are also enriched in the crypts of the small intestine and colon in mammals (*Mao et al., 2015*; *Pédron et al., 2012*; *Saffarian et al., 2017*). Therefore, it will be intertesting to determine whether *Acinetobacter* spp. also modulate EEC function in mammals under high fat diet conditions. Finally, considering the small scale of our mono-assocation screen, we anticipate that additional members of the gut microbiota in zebrafish and other animals will be found to also affect EEC silencing and other aspects of EEC biology.

## Materials and methods

**Key resources table**

| Reagent type (species) or resource | Designation | Source or reference | Identifiers | Additional information |
|---|---|---|---|---|
| Genetic reagent (*D. rerio*) | *TgBAC (neurod1:EGFP)[nl1]* | PMID: 19424431 | | |
| Genetic reagent (*D. rerio*) | *Tg(−5kbneurod1: TagRFP)[w69]* | PMID: 22738203 | | Referred as *Tg(neurod:RFP)* in the paper |
| Genetic reagent (*D. rerio*) | *Tg(sst2:RFP)[gz19]* | PMID: 19281772 | | |
| genetic reagent (*D. rerio*) | *Tg(gcga:EGFP)[ia1]* | PMID: 25852199 | | |
| Genetic reagent (*D. rerio*) | *Tg(−5kbneurod1: Gcamp6f)[icm05]* | PMID: 27231612 | | Referred as *Tg(neurod1:Gcamp6f)* in the paper |
| Genetic reagent (*D. rerio*) | *Tg(−4.5kbfabp2: DsRed)[pd1000]* | PMID: 21439961 | | Referred as *Tg(fabp2:DsRed)* in the paper |
| Genetic reagent (*D. rerio*) | *TgBAC(gata5: lifeAct-EGFP)[pd1007]* | this study | | Generated in this study, Used in *Figures 4*, *5*, *6*, *7*, *8* and *9* |
| Genetic reagent (*D. rerio*) | *Tg(ef1α:xbp1δ-gfp)[mb10]* | PMID: 25892297 | | |
| Genetic reagent (*D. rerio*) | *Tg(NFKB:EGFP)[nc1]* | PMID: 21439961 | | |

*Continued on next page*

*Continued*

| Reagent type (species) or resource | Designation | Source or reference | Identifiers | Additional information |
|---|---|---|---|---|
| Genetic reagent (*D. rerio*) | *Tg(−5kbneurod1: lifeAct-EGFP)*[rdu70] | this study | | Referred as *Tg(neurod1:lifAct-EGFP)* in the paper, Generated in this study, Used in *Figure 4* |
| Genetic reagent (*D. rerio*) | *Tg(−5kbneurod1: Gal4; cmlc2:EGFP)*[rdu71] | this study | | Referred as *Tg(neurod1:Gal4)* in the paper, Generated in this study, Used in *Figure 5* |
| Genetic reagent (*D. rerio*) | *Tg(UAS:Kaede)*[rk8] | PMID: 17406330 | | |
| Genetic reagent (*D. rerio*) | *Tg(ubb:seca5-tdTomato)*[xt24] | PMID: 31391308 | | |
| Genetic reagent (*D. rerio*) | *TgBAC(cd36-RFP)*[pd1203] | this study | | Generated in this study, Used in *Figure 4—figure supplement 4* |
| Genetic reagent (*D. rerio*) | *Tg(tp1bglob: EGFP)*[um14] | PMID: 26153247 | | Referred as *Tg(tp1:EGFP)* in the paper |
| Genetic reagent (*D. rerio*) | *TgBAC(cldn15la: EGFP)*[pd1034] | PMID: 24504339 | | |
| Genetic reagent (*D. rerio*) | *myd88*[b1354] | PMID: 30398151 | | |
| Antibody | anti-PYY (Rabbit Polycolonal) | PMID: 28614796 | | Custom antibody generated in Liddle laboratory, aa4-21 (mouse), IHC (1:100) |
| Antibody | anti-CCK (Goat Polycolonal) | Santa Cruz | Cat# sc-21617, RRID:AB_2072464 | IHC (1:100) |
| Antibody | anti-Sglt1 (Rabbit Polycolonal) | Abcam | Cat# ab14686, RRID:AB_301411 | IHC (1:100) |
| Antibody | GFP (Chicken Polycolonal) | Aves Lab | Cat# GFP-1010, RRID:AB_2307313 | IHC (1:500) |
| Antibody | DsRed (Rabbit Polycolonal) | TAKARA | Cat# 632496, RRID:AB_10013483 | IHC (1:250) |
| Commercial assay or kit | CM-H2DCFDA | Thermofisher | C6827 | |
| Commercial assay or kit | ROS colorimetric assay kit | Sigma | MAK311 | |
| Chemical compond, drug | Phloridzin | Sigma | P3449 | |
| Chemical compond, drug | Thapsigargin | Sigma | T9033 | |
| Chemical compond, drug | Brefeldin A | Sigma | B6542 | |
| Chemical compond, drug | Sodium tauroursodeoxycholic acid | Sigma | T0266 | |

*Continued on next page*

*Continued*

| Reagent type (species) or resource | Designation | Source or reference | Identifiers | Additional information |
|---|---|---|---|---|
| Chemical compond, drug | Orlistat | Sigma | O4139 | |
| Chemical compond, drug | N-acetylcysteine | Invitrogen | C10491 | |
| Chemical compond, drug | N(ω)-nitro-L-arginine methyl ester | Sigma | N5751 | |

## Zebrafish strains and husbandry

All zebrafish experiments conformed to the US Public Health Service Policy on Humane Care and Use of Laboratory Animals, using protocol number A115-16-05 approved by the Institutional Animal Care and Use Committee of Duke University. Conventionally-reared adult zebrafish were reared and maintained on a recirculating aquaculture system using established methods (*Murdoch et al., 2019*). For experiments involving conventionally-raised zebrafish larvae, adults were bred naturally in system water and fertilized eggs were transferred to 100 mm petri dishes containing ~25 mL of egg water at approximately 6 hr post-fertilization. The resulting larvae were raised under a 14 hr light/10 hr dark cycle in an air incubator at 28°C at a density of 2 larvae/mL water. To ensure consistent microbiota colonization, 10 mL filtered system water (5 µm filter, SLSV025LS, Millipore) was added into 3 dpf zebrafish larva that were raised in 25 mL egg water. All the experiments performed in this study ended at 6 dpf unless specifically indicated. The strains used in this study are listed in Key resources table. All lines were maintained on a mixed Ekkwill (EKW) background.

Gateway Tol2 cloning approach was used to generate *neurod1:lifeAct-EGFP* and *neurod1:Gal4* plasmids (*Kawakami, 2007*; *Kwan et al., 2007*). The 5 kb pDONR-neurod1 P5E promoter was previously reported (*McGraw et al., 2012*) and generously donated by Dr. Hillary McGraw. The PME-lifeAct-EGFP (*Riedl et al., 2008*) and the PME-Gal4-vp16 plasmids (*Kwan et al., 2007*) were also previously reported. pDONR-neurod1 P5E and PME-lifeAct-EGFP was cloned into pDestTol2pA2 through an LR Clonase reaction (ThermoFisher, 11791). Similarly, pDONR-neurod1 P5E and PME-Gal4-vp16 was cloned into pDestTol2CG2 containing a *cmlc2:EGFP* marker. The final plasmid was sequenced and injected into the wild-type EKW zebrafish strain and the F2 generation of alleles *Tg(neurod1:lifeAct-EGFP)$^{rdu70}$* and *Tg(neurod1:Gal4; cmlcl2:EGFP)$^{rdu71}$* were used for this study.

The construct used to generate the *TgBAC(gata5:lifeAct-EGFP)* line was made by inserting lifeact-GFP at the *gata5* ATG in the BAC clone DKEYP-73A2 using BAC recombineering as previously described (*Liu et al., 2003*). The BAC was then linearized using I-SceI and injected to generate transgenic lines. Allele *TgBAC(gata5:lifeAct-EGFP)$^{pd1007}$* was selected for further analysis. The construct used to generate the *TgBAC(cd36-RFP)* lines was made by inserting link-RFP before the *cd36* stop codon in the BAC clone DKEY-27K7 using the same BAC recombineering as previously described (*Navis et al., 2013*). Then, Tol2 sites were recombined into the BAC and the resulting construct was injected with transposase mRNA to generate the transgenic lines. Allele *TgBAC(cd36-RFP)$^{pd1203}$* was selected for further analysis.

## Gnotobiotic zebrafish husbandry

For experiments involving gnotobiotic zebrafish, we used our established methods to generate germ-free zebrafish using natural breeding (*Pham et al., 2008*) with the following exception: Gnotobiotic Zebrafish Medium (GZM) with antibiotics (AB-GZM) was supplemented with 50 µg/ml gentamycin (Sigma, G1264). Germ free zebrafish eggs were maintained in cell culture flasks with GZM at a density of 1 larvae/ml. From 3 dpf to 6 dpf, 60% daily media change and ZM000 (ZM Ltd.) feeding were performed as described (*Pham et al., 2008*).

To generate conventionalized zebrafish, 15 mL filtered system water (5 µm filter, SLSV025LS, Millipore, final concentration of system water ~30%) was inoculated to flasks containing germ-free zebrafish in GZM at 3 dpf when the zebrafish normally hatch from their protective chorions. The

same feeding and media change protocol was followed as for germ-free zebrafish. Microbial colonization density was determined via Colony Forming Unit (CFU) analysis. To analyze the effect of high fat feeding on intestinal bacteria colonization, dissected digestive tracts were dissected and pooled (five guts/pool) into 1 mL sterile phosphate buffered saline (PBS) which was then mechanically disassociated using a Tissue-Tearor (BioSpec Products, 985370). 100 μL of serially diluted solution was then spotted on a Tryptic soy agar (TSA) plate and cultured overnight at 30°C under aerobic conditions.

To generate monoassociated zebrafish, a single bacterial strain was inoculated into each flask containing 3dpf germ-free zebrafish. The respective bacterial strain was streaked on a TSA plate and cultured at 28°C overnight under aerobic conditions. A single colony was picked and cultured in 5 mL Tryptic soy broth media shaking at 30°C for 16 hr under aerobic conditions. 250 μL bacterial culture was pelleted and washed three times with sterile GZM and inoculated into flasks containing germ-free zebrafish. OD600 and CFU measurements were performed in each monoassociated culture. The final innoculation density in GZM was $10^8$–$10^9$ CFU/mL. The colonization efficiency was determined at 6 dpf by CFU analysis from dissected zebrafish intestines as described above.

## EEC response assay and image analysis

This assay was performed in *Tg(neurod1:Gcamp6f)* 6 dpf zebrafish larvae. Unanesthetized zebrafish larvae were gently moved into 35 mm petri dishes that contained 500 μL 3% methylcellulose. Excess water was removed with a 200 μL pipettor. Zebrafish larvae were gently positioned horizontal to the bottom of the petri dish right side up carefully avoiding touching the abdominal region and moved onto an upright fluorescence microscope (Leica M205 FA microscope equipped with a Leica DFC 365FX camera). The zebrafish larvae were allowed to recover in that position for 2 min. One hundred μL of test agent was pipetted directly in front of the mouth region without making direct contact with the animal. Images were recorded every 10 s. For fatty acid stimulation, 30 frames (5mins) were recorded. For glucose stimulation, 60 frames (10mins) were recorded. The Gcamp6f fluorescence was recorded with the EGFP filter. The following stimulants were used in this study: palmitic acid/linoleate/dodecanoate (1.6 mM), butyrate (2 mM), glucose (500 mM), fructose (500 mM), galactose (500 mM), cysteine (10 mM). Since palmitic acid/linoleate/dodecanoate was not water soluble by itself, 1.6% BSA was used as a carrier to facilitate solubility. Solutions were filtered with 0.22 μm filter.

Image processing and analysis was performed using FIJI software. The time-lapse fluorescent images of zebrafish EEC response to nutrient stimulation were first aligned to correct for experimental drift using the plugin 'align slices in stack.' Normalized correlation coefficient matching method and bilinear interpolation method for subpixel translation was used for aligning slices (*Tseng et al., 2012*). The plugin 'rolling ball background subtraction' with the rolling ball radius = 10 pixels was used to remove the large spatial variation of background intensities. The Gcamp6f fluorescence intensity in the proximal intestinal region was then calculated for each time point. The ratio of maximum fluorescence ($F_{max}$) and the initial fluorescence ($F_0$) was used to measure EEC calcium responsiveness.

## High fat feeding

The HF feeding regimen was performed in 6 dpf zebrafish larvae using methods previously described (*Semova et al., 2012*). We used an emulsion of chicken egg yolk as our high fat feeding paradigm because it has been used extensively as a high fat diet in zebrafish larvae and adults for metabolic and obesity research (*Carten et al., 2011*; *Maddison and Chen, 2012*; *Minchin et al., 2018*; *Tingaud-Sequeira et al., 2011*; *Zang et al., 2018*; *Zhou et al., 2015*). We refer to this as a high fat meal because lipids comprise greater than 60% dry weight of chicken egg yolk (*Wang et al., 2000*). To perform HF feeding, ~25 zebrafish larvae were transferred into six well plates and 5 mL egg water (for gnotobiotic studies, GZM was used). Replicates were performed in three wells for each treatment group in each experiment. Chicken eggs were obtained from a local grocery store from which 1 mL chicken egg yolk was transferred into a 50 mL tube containing 15 mL egg water (for gnotobiotic studies, sterile GZM was used). Solutions were sonicated (Branson Sonifier, output control 5, Duty cycle 50%) to form a 6.25% egg yolk emulsion. 4 mL water from each well was removed and replenished with 4 mL egg yolk. 4 mL egg water was used to replenish the

control group. The final concentration of egg yolk for HF feeding is 5%. For recovery HF feeding recovery experiments, following 6 or 8 hr of HF feeding, the zebrafish larvae were transferred to a new 6-well plate with clean egg water. Zebrafish larvae were incubated at 28°C for the indicated time. The HF meal was administered between 10am - 12pm to minimize circadian influences. To perform HF feeding in adult zebrafish, 5% egg yolk that is diluted in system water was made similarly as described above. Adult zebrafish raised in the same tank were transferred to 500 mL beakers. For the HF treated groups, the water is removed and 100 mL 5% egg yolk was immediately added to the beaker. For control groups, system water was added to the beaker as a vehicle control.

## Chemical treatment

To block Sglt1, phloridzin (0.15 mM, Sigma P3449) was used to pretreat zebrafish for 3 hr prior to glucose stimulation, and 0.15 mM phloridzin was co-administered with the glucose stimulant solution. To induce ER stress, thapsigargin (0.75 µM, Sigma T9033) and brefeldin A (9 µM, Sigma B6542) were added to egg water and zebrafish were treated for 10 hr prior to performing the EEC activity assay. To block HF meal induced EEC silencing, sodium tauroursodeoxycholic acid (TUDCA; 0.5 mM, Sigma T0266) or orlistat (0.1 mM or 0.5mM, Sigma O4139) were added to the HF meal solution and zebrafish were treated for the indicated time. To block ROS signaling, N-acetylcysteine (NAC, 1 mM, Invitrogen, C10491) or N(ω)-nitro-L-arginine methyl ester (L-NAME, 1 mM, Sigma N5751) were added to the HF meal solution and zebrafish were treated for the indicated time.

## Quantitative RT-PCR

The quantitative real-time PCR was performed as described previously (*Murdoch et al., 2019*). In brief, 20 zebrafish larvae digestive tracts were dissected and pooled into 1 mL TRIzol (ThermoFisher, 15596026). mRNA was then isolated with isopropanol precipitation and washed with 70% EtOH. 500 ng mRNA was used for cDNA synthesis using the iScript kit (Bio-Rad, 1708891). Quantitative PCR was performed in technical triplicate 25 µl reactions using 2X SYBR Green SuperMix (PerfeCTa, Hi Rox, Quanta Biosciences, 95055) run on an ABI Step One Plus qPCR instrument using gene specific primers (*Supplementary file 1*). Data were analyzed with the ∆∆Ct method. *18S* was used as a housekeeping gene to normalize gene expression.

## 16S rRNA gene sequencing

Wild-type adult EKW zebrafish were bred and clutches of eggs from three distinct breeding pairs were collected, pooled, derived into GF conditions using our standard protocol (*Pham et al., 2008*), then split into three replicate flasks with 30 ml GZM as described above. At 3 dpf 12.5 ml 5 µm-filtered system water was inoculated into each flask per our standard conventionalization method. ZM000 feeding and water changes were performed daily from 4 dpf to 5 dpf. At 6 dpf, zebrafish larvae from each flask were divided evenly into a control and a high fat fed group. High fat feeding was performed as described above for 6 hr. Then 1 mL water samples were collected from each flask and snap frozen on dry ice/EtOH bath. For intestinal samples, individual digestive tracts from 6 dpf zebrafish were dissected and flash frozen (3–4 larvae/flask, three flasks/condition). All samples were stored in −80°C for subsequent DNA extraction.

The Duke Microbiome Shared Resource extracted bacterial DNA from gut and water samples using a MagAttract PowerSoil DNA EP Kit (Qiagen, 27100–4-EP) as described previously (*Murdoch et al., 2019*). Sample DNA concentration was assessed using a Qubit dsDNA HS assay kit (ThermoFisher, Q32854) and a PerkinElmer Victor plate reader. Bacterial community composition in isolated DNA samples was characterized by amplification of the V4 variable region of the 16S rRNA gene by polymerase chain reaction using the forward primer 515 and reverse primer 806 following the Earth Microbiome Project protocol (http://www.earthmicrobiome.org/). These primers (515F and 806R) carry unique barcodes that allow for multiplexed sequencing. Equimolar 16S rRNA PCR products from all samples were quantified and pooled prior to sequencing. Sequencing was performed by the Duke Sequencing and Genomic Technologies shared resource on an Illumina MiSeq instrument configured for 150 base-pair paired-end sequencing runs. Sequence data are deposited at SRA under Bioproject accession number PRJNA532723.

Subsequent data analysis was conducted in QIIME2 (*Caporaso et al., 2010*; *Bolyen et al., 2019*). Paired reads were demultiplexed with qiime demux emp-paired, and denoised with qiime dada2

denoise-paired (*Callahan et al., 2016*). Taxonomy was assigned with qiime feature-classifier classify-sklearn (*Pedregosa et al., 2011*), using a naive Bayesian classifier, trained against the 99% clustered 16S reference sequence set of SILVA, v. 1.19 (*Quast et al., 2013*). A basic statistical diversity analysis was performed, using qiime diversity core-metrics-phylogenetic, including alpha- and beta-diversity, as well as relative taxa abundances in sample groups. The determined relative taxa abundances were further analyzed with LEfSe (Linear discriminant analysis effect size) (*Segata et al., 2011*), to identify differential biomarkers in sample groups.

## Immunofluorescence staining and imaging

Whole mount immunofluorescence staining was performed as previously described (*Ye et al., 2015*). In brief, ice cold 2.5% formalin was used to fix zebrafish larvae overnight at 4°C. The samples were then washed with PT solution (PBS+0.75%Triton-100). The skin and remaining yolk was then removed using forceps under a dissecting microscope. The deyolked samples were then permeabilized with methanol for more than 2 hr at −20°C. The samples were then blocked with 4% BSA at room temperature for more than 1 hr. The primary antibody was diluted in PT solution and incubated at 4°C for more than 24 hr. Following primary antibody incubation, the samples were washed with PT solution and incubated overnight with secondary antibody with Hoechst 33342 for DNA staining. The imaging process was performed with a Zeiss 780 inverted confocal and Zeiss 710 inverted confocal microscopes with the 40× oil lenses. The following primary antibodies were used in this study: rabbit anti PYY (custom, aa4-21, 1:100 dilution) (*Chandra et al., 2017*), goat anti-CCK (Santa Cruz SC-21617, 1:100 dilution), rabbit anti-Sglt1 (Abcam ab14686, 1:100 dilution). The secondary antibodies used in this study were from Alexa Fluor Invitrogen. All the secondary antibodies were used at a dilution of 1:250.

To quantify EEC morphology score, chick anti-GFP (Aves GFP1010, 1:500 dilution) and rabbit anti-mCherry (TAKARA 632496, 1:250 dilution) antibodies were used in the fixed *Tg(gata5:lifeAct-EGFP);Tg(neurod1:RFP)* samples to perform immunofluorescence staining. The region following intestine bulb were imaged with a Zeiss 780 inverted confocal and Zeiss 710 inverted confocal microscopes with the 40 × oil lenses. Images were processed with FIJI. The *gata5:lifeAct-EGFP* only stains the apical brush border of the intestine. Total EECs number was assessed via counting RFP+ cell bodies. The number of EECs with intact apical protrusion was assessed via counting the number of RFP+ cells with attachment to GFP staining brush border. EEC morphology for each sample were quantified as ratio between EECs with intact apical protrusion and total EEC number.

For live imaging experiments, zebrafish larvae were anesthetized with Tricane and mounted in 1% low melting agarose in 35 mm petri dishes. The live imaging was recorded with Zeiss 780 upright confocal with a 20 × water lens.

To perform wholemount adult zebrafish intestine imaging in *Tg(neurod1:RFP)*, following indicated treatment, zebrafish was anethetized and the intestine was dissected as described (*Lickwar et al., 2017*). The dissected intestine was immediated fixed in ice cold 4% PFA overnight, and then washed three times with PBS. The proximal intestinal region was dissected and cut open. The flatted intestine tissue was then transferred to glass slides and mounted as described above. The images were obtained using Zeiss 780 inverted confocal 20× dry lens. Three representative regions were image for each fish. The acquired images were processed and analyzed with FIJI software.

To quantify EEC cell volume, the entire pixel volume of *neurod1:RFP* channel in a confocal z-stack was quantified using voxel counter plugin in FIJI software. The entire EEC pixel volume was then divided by EEC number to get the average EEC cell volume in each zebrafish.

## In vivo and in vitro ROS measurement

To measure intestinal ROS production in zebrafish in vivo, zebrafish were incubated with CM-H2DCFDA (0.5 μg/mL, Thermofisher C6827, diluted in gnotobiotic medium) for 1 hr as indicated by previous studies (*Wu et al., 2011*). The zebrafish were then washed with GZM and imaged immediately using an stereofluorescence micropope (Leica M205 FA microscope equipped with a Leica DFC 365FX camera). The mean fluoresence intensity in the proximal intestinal region was quantified using FIJI software. To measure bacterial ROS production in vitro, we used a colorimetric assay kit (Sigma MAK311) as described in previous studies (*Ajiboye et al., 2018*). Briefly, $10^{10}$ log-phase bacteria were harvested and washed with sterile water. The suspended bacteria were then lysed through

three freeze/thaw cycles on dry ice. The remaining debris was pelleted and the supernatant was used for ROS measurement. To measure the effect of high fat condition on bacterial ROS production, $10^{10}$ log-phase bacteria were added to 5 mL 5% chicken egg yolk (diluted in GZM) and cutured at 30°C for 6 hr. ROS measurement was then performed similarly.

## Statistical analyses

The appropriate sample size for each experiment was suggested by preliminary experiments evaluating variance and effects. Using significance level of 0.05 and power of 80%, a biological replicate sample number 10 was suggested for EEC calcium response analysis and a biological replicate sample number 13 was suggested for EEC morphology analysis. For each experiment, wildtype or indicated transgenic zebrafish embryos were randomly allocated to test groups prior to treatment. In some EEC calcium response experiments, less than 10 biological replicate samples were used due to technical limitations associated with live sample imaging. In EEC morphology analysis, each experiment contained 8–15 biological replicates or individual fish samples. Individual data points, mean and standard deviation are plotted in each figure.

The raw data points in each figure are represented as solid dots. The data was analyzed using GraphPad Prism 7 software. For experiments comparing just two differentially treated populations, a Student's t-test with equal variance assumptions was used. For experiments measuring a single variable with multiple treatment groups, a single factor ANOVA with post hoc means testing (Tukey) was utilized. Statistical evaluation for each figure was marked *p<0.05, **p<0.01, ***p<0.001, ****p<0.0001 or ns (no significant difference, p>0.05). Statistical analyses for 16S rRNA gene sequencing data can be found in in the corresponding Materials and methods section above.

## Acknowledgements

We thank Dr Hillary McGraw for the 5 kb pDONR-neurod1 P5E plasmid, Dr Joachim Berger for the pMElifeAct-EGFP plasmid, Dr David Raible for the *Tg(neurod1:RFP)* transgenic line and Dr Claire Wyart for the *Tg(neurod1:Gcamp6f)* transgenic line. We also thank the Duke Light Microscopy Core Facility for equipment access and technical support, the Duke Zebrafish Core Facility for assisting zebrafish husbandry and the Duke Microbiome Shared Resource for 16S rRNA gene sequencing. We are grateful to Colin Lickwar for providing graphic art, and all members of the Liddle and Rawls labs for helpful discussions. This work was supported by grants from the National Institutes of Health R01-DK093399 (to JFR and RAL), R01 DK109368 (to RAL), and R01-DK081426 (to JFR); the Department of Veterans Affairs I01B × 002230 (to RAL); and an Innovation Grant from the Pew Charitable Trusts (to JFR and RAL). LY was supported by the Digestive Disease and Nutrition Training Program at Duke University (T32-DK007568).

## Additional information

### Funding

| Funder | Grant reference number | Author |
|---|---|---|
| National Institute of Diabetes and Digestive and Kidney Diseases | R01-DK093399 | John F Rawls |
| National Institute of Diabetes and Digestive and Kidney Diseases | R01 DK109368 | Rodger A Liddle |
| National Institute of Diabetes and Digestive and Kidney Diseases | R01-DK081426 | John F Rawls |
| Pew Charitable Trusts | | John F Rawls |

The funders had no role in study design, data collection and interpretation, or the decision to submit the work for publication.

## Author contributions
Lihua Ye, Conceptualization, Data curation, Formal analysis, Investigation, Methodology, Project administration; Olaf Mueller, Data curation, Formal analysis; Jennifer Bagwell, Michel Bagnat, Resources; Rodger A Liddle, John F Rawls, Conceptualization, Resources, Supervision, Funding acquisition, Project administration

## Author ORCIDs
Lihua Ye (ID) https://orcid.org/0000-0001-6790-9743
Michel Bagnat (ID) http://orcid.org/0000-0002-3829-0168
John F Rawls (ID) https://orcid.org/0000-0002-5976-5206

## Ethics
Animal experimentation: All zebrafish experiments conformed to the US Public Health Service Policy on Humane Care and Use of Laboratory Animals, using protocol number A115-16-05 approved by the Institutional Animal Care and Use Committee of Duke University.

## Decision letter and Author response
Decision letter https://doi.org/10.7554/eLife.48479.sa1
Author response https://doi.org/10.7554/eLife.48479.sa2

# Additional files

## Supplementary files
• Supplementary file 1. Sequence of the primers that were used for real-time quantitative PCR.
• Supplementary file 2. Amplicon sequence variant table of 16S rRNA gene sequencing analysis.
• Supplementary file 3. LEfSe analysis of relative taxa abundance in sequenced media samples.
• Supplementary file 4. LEfSe analysis of relative taxa abundance in sequenced gut samples.
• Transparent reporting form

## Data availability
Sequencing data have been deposited at SRA under Bioproject accession number PRJNA532723. All data generated or analyzed during this study are included in the manuscript and supporting files. Source data files have been provided for Figures 1–9, Figure 2—figure supplement 1. The link for accessing the source data is https://doi.org/10.5061/dryad.mb004d1.

The following datasets were generated:

| Author(s) | Year | Dataset title | Dataset URL | Database and Identifier |
|---|---|---|---|---|
| Lihua Ye, Olaf Mueller, Jennifer Bagwell, Michel Bagnat, Rodger A Liddle, John F Rawls | 2019 | Impact of a high-fat meal on the gut microbiota in zebrafish larvae | https://www.ncbi.nlm.nih.gov/bioproject/PRJNA532723/ | NCBI, PRJNA532723 |
| Rawls J | 2019 | Data from: High fat diet induces microbiota-dependent silencing of enteroendocrine cells | https://doi.org/10.5061/dryad.mb004d1 | Dryad Digital Repository, 10.5061/dryad.j1fd7 |

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
