## [Decision Letter]

**Acceptance summary:**

The paper reports a mature and detailed morphological and functional study of enteroendocrine cells in zebrafish larvae, using strain combinations with reporter systems, intestinal nutrient challenges, and microbiota manipulation to show that 'high fat diet' induces microbiota-dependent silencing of enteroendocrine cells. The silencing of the reporter Gcamp6f calcium response to palmitate or glucose stimulation is shown to be associated with morphological cellular apical extension loss during the high-fat pre-feeding and with ER stress.

**Decision letter after peer review:**

[Editors’ note: the authors were asked to provide a plan for revisions before the editors issued a final decision. What follows is the editors’ letter requesting such plan.]

Thank you for sending your article entitled "High fat diet induces microbiota-dependent silencing of enteroendocrine cells" for peer review at *eLife*. Your article is being evaluated by Wendy Garrett as the Senior Editor, a Reviewing Editor, and three reviewers.

Given the list of essential revisions, including new experiments, the editors and reviewers invite you to respond within the next two weeks with an action plan and timetable for the completion of the additional work. We plan to share your responses with the reviewers and then issue a binding recommendation.

The reviewers have discussed the paper after submitting their critiques entirely independently. The consensus is that the major concern relates to physiological relevance due to a potentially acute stress artefact caused by either (1) the direct action of toxic lipid intermediates, and (2) the indirect action of cellular volume dysregulation. Your manuscript shows that microbial processing of lipids is a critical component in the desensitization process, and this could implicate beta-oxidation products, or other oxidative signals that directly target Ca channel and G-protein receptor function via adduct formation; e.g. nitrosative and oxidative forms of lipids. Further, Gcamp is known to impart sensitivity to cellular/ER stress. We think that this general concern needs to be experimentally addressed, and we would be grateful for your responses about this before proceeding.

*Reviewer #1:*

The paper reports a mature and very detailed morphological and functional study of enteroendocrine cells in zebrafish larvae, using strain combinations with reporter systems, intestinal nutrient challenges, and microbiota manipulation to show that 'high fat diet' induces microbiota-dependent silencing of enteroendocrine cells. By this, the authors mean that the egg yolk preparation exerts these effects in mice with associated microbes, and that the EEC morphology score does not change when the egg yolk manipulation is carried out in germ-free larvae. The silencing of the reporter Gcamp6f calcium response to palmitate or glucose stimulation is nicely shown to be associated with morphological apical extension loss during the high-fat pre-feeding and with ER stress.

There is a considerable body of strongly supportive data in the paper, including Kaede system consideration of neogenesis, xbp1 readouts of ER stress, and monocolonization to identify a taxon (Acinetobacter: likely aligned with the beta-oxidative pathway) that can phenocopy the morphological result of high fat preconditioning.

1) Since palmitate preconditioning alone does not phenocopy the morphological score or the response to subsequent glucose stimulation, notwithstanding the result of orlistat pretreatment, it seems unlikely that the effect would be abrogated by blockade of carbonic acid receptors alone. Given the results with Acinetobacter, I wondered (i) whether oxidative stress (generated secondarily through the microbiota through aerobic metabolism and iron uptake e.g. PMID 29614366) would be playing a role here and (ii) to what extent the response can really be considered physiological. For example, can the dynamic morphological changes be seen at a non-larval stage depending on diet in zebrafish or in mammals? These are my two top concerns for the significance of the elegant work in the paper. I worry that the claims of a new physiologically dynamic EEC pathway in the discussion may be missing a stress response.

2) I think that it would be asking too much to carry out further experiments with strain combinations of chemoreceptor-deficient larvae – especially as the initial effect may be unrelated to chemoreceptor signaling as above. Nevertheless, egg yolk provides a high but not exclusive lipid load, making the title, abstract and interpretations throughout the text inexact.

3) I have some questions about the images presented. For example, for the Gcamp6f calcium response in Figure 5E there seems to be a response at F0 and there appears to be clear fluorescence distally. In the xbp1 response in Figure 5K, K', only 1 (perhaps 2) cells seem to be to be convincingly positive, making one uncertain about the variable results in panel O (perhaps the ER stress is transitory?). In panels A and B, I was also unsure about the precise meaning of 'n=4, each replicate from 20 pooled fish sample(s), 3 technique (technical?) replicates': could the points be shown. (Note that there are a series of typos further in the legend to Figure 5).

4) It is rather strange to present the EEC calcium response in different ways in two figures (Figure 7—figure supplement 1 and Figure 8—figure supplement 1) that should be comparable. It is not clear what the error bars in Figure 7—figure supplement 1D represent, and the points for the controls in 11A are invisible.

5) In Figure 7I, since the groups GF and CV appear not to overlap, lack of significance presumably reflects lack of power.

6) The different filters in the subpanels of the figures could be specified in the legends for clarity.

*Reviewer #2:*

This exciting study in a zebrafish model demonstrates high fat diet induced silencing of enteroendocrine cell (EEC) function that is mediated in a microbiota-dependent fashion. This important work represents a tour-de-force experimental account of EEC subtype distribution and function in the zebrafish, visualized as in vivo nutrient sensing in real time using the rapid genetic intracellular Ca^2+^ indicator Gcamp6F expressed selectively in EEC. Fatty acids palmitate, linoleate and dodecanoate rapidly induce intracellular Ca^2+^ transients in proximal EEC expressing PYY and CCK, which are evolutionary conserved hormones required for energy homeostasis and regulation of satiety. A high fat diet enriched in egg yolk desensitizes intracellular Ca^2+^ transients in these ECC subtypes and the process is shown to be critically dependent on host lipase activity, sodium-dependent glucose transport and microbiota composition, with Acinetobacter-dominating communities driving the physiology. The EEC desensitization process to fatty acid signaling was additionally shown to be associated with morphological withdrawal of apical EEC processes into a closed cellular configuration that is no longer capable of surveillance of the gut lumen, a term coined as EEC silencing and involves ER stress. The authors describe this as a new microbiota-dependent mechanism where a high fat diet can uncouple nutrient sensing and energy homeostasis in the host, as well as desensitize satiety signals that control food intake. These observations may extend to mammalian pathophysiology, especially since a high fat diet is often associated with enriched proteobacteria communities (including Acinetobacteria spp) that colonize the mucosa and can invade the intestinal crypts where EEC populations often dominate. The potentially important study conclusions would benefit from clarifying the following points:

1) The short chain fatty acid butyrate did not induce any observed Ca^2+^ transients which is contrary to findings in mammals e.g. serotonin-release by EEC. Bile acids also signal via this microbiota-dependent mechanism in mammals, but this is not explored or discussed in the current work. It is expected that bile acid profiles will change drastically in this model in response to dietary fat challenge, and some data are presented to support a bile acid dependent response. In view that many secondary bile acids are potent G-coupled receptor ligands that induce Ca^2+^ transients and can desensitize related signaling pathways, some consideration of the potential role of bile acids and the apparent disconnect to mammalian physiology in the discussion is merited.

2) Acute toxicity of microbial processed metabolites is not ruled out as the major effector. Is a basolateral or whole body stimulus still possible in these EEC to show that they remain viable cells? There is no direct evidence to rule out toxicity since apoptosis is a late stage indicator. Do the conventional animals survive longer term on this artificially high fat diet? Only a 6 hour time point is shown. Is the EEC desensitization process reversed? How does this acute fat intake relate to the mammalian diet since this could have more acute toxicity effects? The dietary fat composition used should be better justified and explained in this context.

3) It is not clear that the ER stress response is functionally coupled to EEC hormone physiology in this model. The ER stress response may reflect that experiments were done in GCaMP6, rather than wildtype animals. GCaMP proteins buffer intracellular calcium levels and it is possible that EEC could be more vulnerable to ER stress in this context and lipid exposure in this case is a second hit.

4) In the figures and videos, the high fat diet appears to trigger a distended GI that could be involved in the EEC silencing. Since SGLT-1 is a known cell volume regulator and inhibits the EEC signaling in the same manner, some data to rule out abnormal cell volume regulation in this model seems important. This is highly relevant in the context of store-operated (Orai-STIM) and/or TRP channels which couple apical plasma membrane to the ER and are primary regulators of Ca^2+^ transients associated with cell volume regulation.

*Reviewer #3:*

In this manuscript, Ye et al., establish and validate a novel zebrafish reporter of enteroendocrine cell (EEC) activation and investigate the effects of high fat feeding on EEC morphology and function. Their main finding is that high fat diet (HFD) induces a silencing program in EECs, which is reflected by two phenotypical observations: (1) reduced response of EECs to fatty acid or glucose stimulation and (2) changes in cell shape, i.e. loss of apical extensions. Given the similarity of EECs to neurons, a functional "desensitization" of EECs after continuous stimulation (by HFD) is not unexpected. The observed plasticity of EECs to convert from an open to a closed state, however, is highly interesting and in contrast to previous speculations that "open" and "closed" EECs may represent two independently differentiated EECs subsets. Mechanistically, Ye at al., demonstrate critical roles of ER stress, lipase activity and colonization by microbiota in HFD-induced EEC silencing.

Overall, the present study is novel, well executed and elegantly presented. All major morphological observations were quantified appropriately. Raw data together with results from the performed statistical testing were made publicly available by the authors to support reproducibility.

Essential revisions:

1) Given the observed silencing of EECs after HFD, the finding of increased transcription of several genes encoding for EEC hormones (Figure 5A) is counter-intuitive and suggests a translational block and/or impairment of vesicle excretion. It would be of value to discuss this.

2) The absence of a significant increase of Xbp1t and Xbp1s after HFD (Figure 5B) is surprising in light of the observed increase in Grp78 (Bip) and the reported findings from the Xbp1s reporter model. This discrepancy might be due to a dilution effect, as whole tissue was analyzed in Figure 5B. Although not critical, it would be interesting to see the results of Xbp1 splicing assays of these samples (i.e. electrophoresis of amplified Xbp1 transcripts).

3) Thapsigargin induces ER stress via specific inhibition of sarco-endoplasmic reticulum Ca^2+^-ATPases, which subsequently raises cytosolic calcium. This is an important confounding variable when using a calcium reporter, as it is done here. It would be critical to provide raw fluorescence intensities for Figure 5H and I (as done in Figure 2D) as the reporter signal may already be saturated at baseline. Alternatively, the authors could provide functional data for the BFA or TUDCA experiments as these compounds should not affect calcium signaling.

4) Pre-treatment of animals with palmitate (instead of HFD) reduced the ability of EECs to respond to subsequent palmitate stimuli but did not induce the morphological changes seen after HFD (Figure 7—figure supplement 1). This unexpected finding, which suggests the presence of at least two independent mechanisms of EEC silencing, is attributed to "undefined signals from fat digestion" (subsection “The effects of diet and microbes on EEC silencing”), which is too vague in my eyes. The suggested role of mechanical pressure on EECs by adjacent enterocytes that have accumulated lipid droplets is not supported by experimental data. Is glycerol thought to play a role? Is the length/nature of fatty acids important? Does the palmitate diet increase EEC ER stress? It would valuable to at least discuss the former two points and to experimentally address the latter one.

5) The authors convincingly demonstrate that mono-colonization of animals with Acinetobacter sp. ZOR0008 is sufficient to permit HFD-induced changes of EEC morphology. It remains unclear, however, whether this observation is accompanied with a reduced EEC response to fatty acids and/or glucose. This is a challenging experiment as it would require the generation of germ-free neurod1:Gcamp6f fish and is not absolutely critical. Nonetheless – if feasible – these experiments would significantly strengthen the manuscript given the observation that morphologic and functional EEC silencing can be uncoupled in certain conditions as described in point 4.

[Editors’ note: formal revisions were requested, following approval of the authors’ plan of action.]

---

## [Author Response]

[Editors’ notes: the authors’ response after being formally invited to submit a revised submission follows.]

The reviewers have discussed the paper after submitting their critiques entirely independently. The consensus is that the major concern relates to physiological relevance due to a potentially acute stress artefact caused by either (1) the direct action of toxic lipid intermediates, and (2) the indirect action of cellular volume dysregulation. Your manuscript shows that microbial processing of lipids is a critical component in the desensitization process, and this could implicate beta-oxidation products, or other oxidative signals that directly target Ca channel and G-protein receptor function via adduct formation; e.g. nitrosative and oxidative forms of lipids. Further, Gcamp is known to impart sensitivity to cellular/ER stress. We think that this general concern needs to be experimentally addressed, and we would be grateful for your responses about this before proceeding.

We are grateful to you and the three reviewers for their very positive evaluation of our manuscript. We appreciate and understand the concerns highlighted above as well as those itemized below. Before we provide itemized responses to each of the reviewers’ concerns below, we will briefly address the three major concerns mentioned in the editor’s introduction above: (1) the potential for direct action of toxic lipid intermediates, (2) the potential for indirect action of cellular volume dysregulation, and (3) the potential ability of Gcamp to impart sensitivity to cellular/ER stress. Whereas concern (3) would indeed represent a problematic stress artefact, we feel our existing data already address this; and we think concerns (1) and (2) do not represent “acute stress artefacts” but instead represent two of several potential physiologically-relevant mechanisms underlying this EEC silencing phenotype. However, in response to these reviewer concerns, we have now conducted new experiments that tend to exclude concerns (1) and (2) as potential mechanisms.

1) We agree that microbe-induced ROS, lipid peroxidation, and/or beta-oxidation products are potential candidates that may lead to the EEC silencing phenotype. We would first like to clarify what appears to be some confusion about the interpretation of our results. The comments in the editor’s introduction state that our “manuscript shows that microbial processing of lipids is a critical component in the desensitization process”, and that particular interpretation appears to have raised reviewer interest in the potential role of ROS, lipid peroxidation, or beta-oxidation products. We agree that those are potential candidate mechanisms that may lead to the EEC silencing phenotype, however we respectfully point out that none of our current data suggest that microbial processing of lipids directly contribute to this phenotype. Our data do show that EEC silencing is induced by a high-fat (HF) meal, and that phenotype requires the presence of complex microbiota or a specific Acinetobacter member. But we still do not know the mechanism by which microbial colonization promotes this EEC silencing phenotype. Besides microbe-induced ROS, lipid peroxidation, or beta-oxidation products, we feel there are several other potential mechanisms that may be underlying this phenotype (e.g., microbial lipase production or emulsification activity, microbial regulation of host lipase activities, microbial metabolism of bile acids, microbial stimulation of the host immune system, microbial influence on intestinal gene expression, etc.). We further address this reviewer/editor concern below, and we have conducted new experiments to test whether microbial ROS production/lipid peroxidation may directly contribute to the EEC silencing phenotype. The results of those experiments suggest that these mechanisms do *not* contribute to EEC silencing, so we hope this reviewer concern is now resolved.

2) We agree that changes in cell volume are a potential physiological consequence of HF meal digestion and could contribute to the observed EEC silencing phenotype. The reviewer suggests signaling pathways that might alter EEC cell volume. We further address this reviewer/editor concern below, and we have conducted new measurements to test the extent to which EEC cell volume changes during HF feeding. The results of those experiments suggest that EEC silencing is *not* associated with significant changes in EEC cell volume, so we hope this reviewer concern is now resolved.

3) We are aware of the caveats of Gcamp mentioned above. For this very reason, multiple experiments already included in the submitted manuscript use zebrafish that do not express Gcamp yet still show the same phenotype of HF feeding induced EEC silencing (Figure 4, Figure 6L-Q, Figure 7G-P, Figure 8A-B, I-K, Figure 9H and Figure 4—figure supplement 1A-C, Figure 4—figure supplement 4, Figure 4—figure supplement 5, Figure 6—figure supplement 1 and Figure 9—figure supplement 3A-D). Below we proposed additional experiments that could be performed to further demonstrate this.

Below we provided responses to each of the reviewers’ points and itemized potential experiments we could do to address each major reviewer concern. These experiments are listed below their respective reviewer concern but labeled as experiments (a) – (r) below to facilitate discussion (Author response table 1). Though we originally identified 19 different experiments that could be done, there were 9 experiments that we felt would address the major two points mentioned above and also improve the manuscript in important ways. These 9 experiments are a,b,c,e,i,k,n,q,r. In our revision plan, we proposed to do these 9 experiments for our revised manuscript, and this plan was deemed acceptable by the editors.

**Author response table 1. resptable1:** 

	Experiment	Time required	Timeline
a	EEC silencing-ROS inhibitor	3 weeks	
b c	in vitro bacteria ROS measurement in vivo ROS measurement	2 weeks 2 weeks 2 weeks	Month 1
d	in vitro lipid peroxide
e	Adult fish EEC silencing
f	GF, CV basal EEC morphology
g	SCFA		
h	TGR5	2 weeks	
i	EEC silencing reversal
j	Gcamp-ER stress
k	EEC volume measurement	3 weeks 2 weeks	Month 2
l m	Orai-Stim inhibitor TRPM inhibitor
n	Temporal ER stress measurement
o	FACS-EEC qPCR
p	BFA EEC calcium response	2 weeks	
q	Palmitate ER stress
r	Acinetobacter EEC calcium	3 weeks	

We have now performed all 9 experiments (a,b,c,e,i,k,n,q,r). These data are displayed in the new Figure 5, Figure 4—figure supplement 1, Figure 4—figure supplement 3D, Figure 4—figure supplement 5, Figure 6—figure supplement 2, Figure 7—figure supplement 1F-H, Figure 9—figure supplement 2 and Figure 3—figure supplement 3. We have updated the associated text in the sections of Results section, Discussion section and Materials and methods section in the revised manuscript.

In summary, our new data suggest that:

(1) Microbe-induced ROS, lipid peroxidation, and/or beta-oxidation products are unlikely to be the mechanism that leads to the EEC silencing phenotype (see detailed explanation below and results in the Figure 9—figure supplement 3).

(2) EEC silencing is a physiologically relevant response that can also be observed in adult zebrafish. We also find that EEC silencing is reversible, and not associated with change in EEC cell volume. Together these new data demonstrate that EEC silencing is an active and reversible cellular adaptation to high fat meal and is not an acute stress artifact on EECs or the animal.

(3) Palmitate treatment alone does not induce ER stress in EECs.

(4) Mono-association of germ-free zebrafish with *Acinetobacter* sp. is sufficient to induce both major aspects of EEC silencing following HF feeding.

Below we provide itemized responses to each of the reviewers’ concerns:

Reviewer #1:1) Since palmitate preconditioning alone does not phenocopy the morphological score or the response to subsequent glucose stimulation, notwithstanding the result of orlistat pretreatment, it seems unlikely that the effect would be abrogated by blockade of carbonic acid receptors alone. Given the results with Acinetobacter, I wondered (i) whether oxidative stress (generated secondarily through the microbiota through aerobic metabolism and iron uptake e.g. PMID 29614366) would be playing a role here and (ii) to what extent the response can really be considered physiological. For example, can the dynamic morphological changes be seen at a non-larval stage depending on diet in zebrafish or in mammals? These are my two top concerns for the significance of the elegant work in the paper. I worry that the claims of a new physiologically dynamic EEC pathway in the discussion may be missing a stress response.

We appreciate the reviewer’s comments and suggestions. As an aside, we presume the reviewer meant “fatty acid receptor” and not “carbonic acid receptor” above. The reviewer raises an interesting idea that HF-induced EEC silencing is mediated by oxidative stress induced by microbiota. This idea is further expanded in the editor’s introduction above, raising the idea that EEC silencing might be caused by lipid peroxidation and the resulting toxic lipid intermediates, which could be induced by bacterial FAO or ROS. As described above in our opening remarks to the editor, we don’t think our data provide specific support for this model, but we agree that this is one of several potential mechanisms underlying EEC silencing. One prediction of this model is that HF meal feeding would induce oxidative stress response pathways in the host in colonized zebrafish at higher levels than germ-free controls. On one hand, our lab’s previous studies indicate that there is greater NF-κB activity in conventionalized (CV) zebrafish than germ-free (GF) zebrafish (PMID: 21439961). Considering NF-κB is one of the main host transcriptional pathways toward increased ROS production (PMID: 16317160, PMID: 16723122), this finding may lead to the hypothesis that higher ROS is produced in CV zebrafish than GF zebrafish. However, our lab’s additional unpublished data using CM-H_2_DCFDA (a molecular probe that indicates ROS levels in vivo; Author response image 1) shows no significant differences between GF and CV zebrafish larvae at baseline without HF meal (with or without PMA induction).

**Author response image 1. respfig1:** CM-H_2_DCFA (a molecular probe indicating ROS levels) shows highest activity in the gut lumen, gall bladder, gut epithelium, and nephric duct in 6dpf conventionally-raised zebrafish larvae (left panel). Platereader measurements of CM-H_2_DCFA levels in GF and CV zebrafish larvae with or without PMA induction, show no significant differences (right panel).

Although we feel these data somewhat diminish support for the reviewer’s hypothesis that ROS and resulting toxic lipid intermediates are induced by microbiota, we offered several potential experiments below that could further explore this working. Note that the lack of genetic tools for our Acinetobacter strain limits our ability to test this working hypothesis via bacterial genetic analysis within a reasonable timeframe.

(a) Attempt to rescue the EEC silencing phenotype in colonized zebrafish larvae with a ROS scavenger such as glutathione and DMTU.

(b) Measure Acinetobacter ROS production with and without HF meal stimulation in vitro using a CellROX or peroxide assay kit.

(c) Measure in vivo intestinal luminal ROS levels in GF, CV and Acinetobacter monoassociated zebrafish larvae following HF feeding using CM-H2DCFDA.

(d) Compare lipid peroxidation capabilities between Acinetobacter and other tested bacterial strains in vitro using the TBARS assay.

The results for experiments a-c are displayed in Figure 9—figure supplement 3. These new data are reported in the subsection “High fat feeding induces EEC silencing in a microbiota dependent manner” and subsection”. The effects of diet and microbes on EEC silencing”. We found that inhibition of ROS signaling through ROS inhibitors like N-acetylcysteine failed to prevent the EEC silencing phenotype in vivo. Further, neither colonization with microbiota nor HF feeding led to elevation in ROS levels as measured by CM-H2DCFDA fluorescence. Finally, *Acinetobacter* sp. does not produce detectable ROS with or without HF stimulation in vitro. These data suggest that HF feeding and microbiota colonization induce EEC silencing likely occurs through an ROS independent mechanism. We have added these new results at the very end of the Results section, but if the reviewers would prefer we remove these negative data from the manuscript, we’d be happy to do that instead.

The reviewer’s general remarks above also mention the possibility that Acinetobacter might induce EEC silencing somehow through beta-oxidation, but that does not appear to be a major concern, so we have not addressed it in our response. If the reviewer is in fact interested in the potential role of lipid intermediates produced by bacterial FAO on EEC silencing (independent of ROS), we have no data to address this and we would not be able to experimentally test this with new experiments in a timely manner. Though it would be potentially interesting to know if there were significant effects of the HF meal and/or microbiota on the diversity and abundance of FAO-derived lipid intermediates in larval zebrafish digestive tracts, such assays have not been developed in our lab and we are concerned that we would not easily be able to assign those differences to microbial vs host FAO. The only bacterium we have identified that induces EEC silencing is this *Acinetobacter* isolate from the zebrafish intestine which has no established system for genetic manipulation, so genetic approaches to test the requirement for *Acinetobacter* FAO on EEC silencing are not currently feasible. Further, we are unaware of any way to pharmacologically inhibit FAO in Acinetobacter (or other bacteria) that would not also affect host FAO.

We appreciate the reviewer’s comment regarding physiological relevance, and we agree that it would be helpful to address this by performing similar experiments in adult zebrafish or a mammalian system. We performed our studies in larval zebrafish because of their small size and transparency which makes the in vivo calcium imaging and whole-mount immunofluorescence imaging of the intestine feasible. Both are very challenging to achieve in adult zebrafish or mouse. Because EECs do not reside in the same focal plane in the intestine, it is almost impossible to recover and quantify EEC morphology using classic tissue sectioning approaches. Although 3D immunofluorescence imaging is a possibility (PMID: 23936537), quantitative analysis using this technique has not been established. Therefore, though we have already given this issue a lot of consideration, we unfortunately do not have a simple way to fully reproduce our studies on EEC morphology in adult zebrafish or a mammal. However, we proposed the following experiment in attempt to address this reviewer concern:

(e) Using some of the transgenic lines used here, feed the HF meal to adult zebrafish and attempt to quantify EEC morphology and nutrient sensitivity to see if EEC silencing is induced.

We performed the indicated experiment (e) to quantify the effect of HF feeding on adult zebrafish EEC morphology. The results are displayed in Figure 4—figure supplement 5. These experimental data are explained in the subsection “High fat feeding induces morphological adaption in enteroendocrine cells”. Similar to our findings in larvae zebrafish, HF feeding also significantly induced more “closed” type EEC morphology in adult zebrafish, indicating that the HF feeding induced EEC silencing is a physiological relevant adaptation that can occur at multiple life stages. We do not currently have methods to permit evaluation of nutrient sensitivity in adult zebrafish EECs, but we feel this morphologic data from adult zebrafish largely addresses the reviewer concern.

2) I think that it would be asking too much to carry out further experiments with strain combinations of chemoreceptor-deficient larvae – especially as the initial effect may be unrelated to chemoreceptor signaling as above. Nevertheless, egg yolk provides a high but not exclusive lipid load, making the title, abstract and interpretations throughout the text inexact.

We agree with the reviewer that a definitive genetic/chemical test of all potential receptors for fatty acids and other lipids and candidate signaling pathways is beyond the scope of this work. We adopted egg yolk diet as our HF feeding paradigm because chicken egg yolk is the most used HF diet in zebrafish larvae and juveniles for the metabolism and obesity research (PMIDs: 30177968, 22721970, 21724975, 25497901, 26607039). We also agree with the reviewer that the egg yolk meal used here provides a high, but not exclusive lipid load. However, we feel that our use of the term “high-fat meal” is consistent with the standards in the metabolism field – wherein diets are often referred to in manuscripts as “high-fat” or “high-carb” when in fact there are other nutrient types present, and also when the increased abundance of fat in a “high-fat” diet means relative abundance of other nutrient type(s) is being reduced. However, that reduction is seldom mentioned in the diet name throughout a paper. So, with that in mind, we would like to continue to call this a “high-fat meal” in the paper, since we feel it is accurate and within field norms. To help address the reviewer’s concern, we have added text to the subsection “EEC physiology in zebrafish” and subsection “High fat feeding” that clarifies the lipid content of chicken egg yolk (60%), and provides citations to papers that describe contents of egg yolk and extensive use of egg yolk as a high fat diet in zebrafish. We hope this satisfies the reviewer’s concern.

3) I have some questions about the images presented. For example, for the Gcamp6f calcium response in Figure 5E there seems to be a response at F0 and there appears to be clear fluorescence distally. In the xbp1 response in Figure 5K, K', only 1 (perhaps 2) cells seem to be to be convincingly positive, making one uncertain about the variable results in panel O (perhaps the ER stress is transitory?). In panels A and B, I was also unsure about the precise meaning of 'n=4, each replicate from 20 pooled fish sample(s), 3 technique (technical?) replicates': could the points be shown. (Note that there are a series of typos further in the legend to Figure 5).

We have attempted to address these reviewer concerns with the following figure and text edits:

Figure 5E is now updated to Figure 6E: We agree that this Gcamp fluorescence in the distal gut is unexpected. We confirm this was observed in almost all animals that received that treatment. In this experiment, we pretreated the zebrafish with thapsigargin to induce ER stress. It is therefore possible that thapsigargin affects EECs in the distal intestine which altered the F0 fluorescence here. We have not explored this further in the present manuscript. However, we note that treatment with thapsigargin did not alter Gcamp basal fluorescence intensity in the proximal intestine (Figure 6—figure supplement 2A-C), the primary region where EECs respond to palmitate and glucose stimulation.

Figure 5K and K’ is now updated to Figure 6M, M’: When we quantified s-xbp1+(GFP+) expression, we obtained a Z-stack of the whole intestine and quantified the s-xbp1+ cells in the whole z-stack field. In presenting the image, we used the focal plane from the z-stack to clearly show the overlapping of s-xbp1+ cells with the EEC marker. In response to this reviewer concern, we have now used confocal z-stack projection instead of single confocal plan (Figure 6L-M) to clearly display the induction of s-xbp1+ EECs following HF feeding. In addition, we now provide z-stack videos from control and HF fed zebrafish intestines in Video 7 and Video 8.

Figure 5A, B is now updated to Figure 6A, B: “n=4, each replicate from 20 pooled fish sample(s), 3 technique (technical?) replicates.” We are sorry for the confusion. This means that we pooled 20 fish to extract RNA. For both control and HF fed larvae, we have four groups of pooled fish. When we performed the RT-PCR assays, we used 3 technical replicates and used their mean for quantification. We have now changed Figure 5 (Current Figure 6) legend and the associated method section to clarify this. In addition, we have changed the Figure 6A, B format to a bar graph including individual points. We have also corrected typographical errors and updated Figure 5 (current Figure 6) legend.

4) It is rather strange to present the EEC calcium response in different ways in two figures (Figure 7—figure supplement 1 and Figure 8—figure supplement 1) that should be comparable. It is not clear what the error bars in Figure 7—figure supplement 1D represent, and the points for the controls in 11A are invisible.

We thank the reviewer for pointing this out, and we have corrected this. We have now changed the Figure 7—figure supplement 1D to a bar graph with individual points similar to that of Figure 8—figure supplement 1A-B. The error bars in Figure 7—figure supplement 1D represent the variation for Gcamp signals in different biological replicate samples. We have now changed the color of Figure 8—figure supplement 1A to make the points more visible.

5) In Figure 7I, since the groups GF and CV appear not to overlap, lack of significance presumably reflects lack of power.

Noted that Figure 7I is now updated to Figure 8I. We agree with the reviewer’s assessment, but we posit that the presence or absence of statistical significance in that particular comparison does not significantly impact our overall conclusions. We could do the following experiment:

(f) perform a new experiment including a larger number of biological replicates to compare EEC morphology score in CV and GF groups.

6) The different filters in the subpanels of the figures could be specified in the legends for clarity.

We thank the reviewer for pointing this out – we have specified the subpanels in the all the figure legends.

Reviewer #2:[…] The potentially important study conclusions would benefit from clarifying the following points:1) The short chain fatty acid butyrate did not induce any observed Ca^2+^ transients which is contrary to findings in mammals e.g. serotonin-release by EEC. Bile acids also signal via this microbiota-dependent mechanism in mammals, but this is not explored or discussed in the current work. It is expected that bile acid profiles will change drastically in this model in response to dietary fat challenge, and some data are presented to support a bile acid dependent response. In view that many secondary bile acids are potent G-coupled receptor ligands that induce Ca^2+^ transients and can desensitize related signaling pathways, some consideration of the potential role of bile acids and the apparent disconnect to mammalian physiology in the discussion is merited.

The lack of butyrate response was also somewhat surprising to us, however, a zebrafish short-chain fatty acid (SCFA) receptor has not yet been identified. We expect such lack of response may be due to either lack of conservation in a short-chain fatty acid receptor gene or zebrafish EECs do not express said a short-chain fatty acid receptor. In addition, it remains unknown whether zebrafish larval microbiota are even capable of producing SCFAs, especially at these early life stages. If this question is of significant concern to the reviewer, one experiment we could do is the following:

(g) We could test the ability of other SCFAs including acetate and propionate to induce the EEC calcium response in zebrafish larvae. However, due to the current lack of understanding of zebrafish SCFA biology we will be limited in how far we can extend these studies beyond that point.

We agree with the reviewer that bile acids may be playing a role in this process considering its important role in lipid digestion. We also appreciate that bile acids could directly activate EECs possibly through TGR5. So far, very limited research has been done on zebrafish bile (see PMID 20113173), with only a single bile salt species having been reported in zebrafish (5-α cyprinol sulfate), no reported efforts to search for other potential primary and secondary bile acid species, and no reported efforts to define bacterial species from the zebrafish intestine capable of bile acid modifications. So, while we view this as an attractive potential mechanism, we have very little foundational knowledge or reagents to explore this hypothesis at this time. To explore the reviewer’s idea further, we have queried an unpublished single-cell RNA-seq dataset in our lab from intestinal epithelial cells from zebrafish larvae. We find that Tgr5 is expressed at a low level in one subtype of EECs, but not in other EEC subtypes (unpublished data not shown). The fact that Tgr5 is expressed in only a subset of EECs reduces the likelihood that it explains the generalized phenotype of EEC silencing which occurs in most/all EECs in the proximal intestine. We do have future plans to conduct a more detailed analysis of host/microbial metabolism and diversity of bile acids in the zebrafish. At this time, we would only be able to conduct the following experiments to explore the Tgr5 hypothesis specifically at this time, if it is of sufficient importance to the reviewer:

(h) We could test the role of Tgr5 in EEC silencing by (i) attempting to rescue the EEC silencing phenotype with triamterene, a known TGR5 inhibitor, or (ii) attempt to induce the EEC silencing phenotype with the Tgr5 agonist 3-aryl-4-isoxazolecarboxamides (PMID: 19902954).

2) Acute toxicity of microbial processed metabolites is not ruled out as the major effector. Is a basolateral or whole body stimulus still possible in these EEC to show that they remain viable cells? There is no direct evidence to rule out toxicity since apoptosis is a late stage indicator. Do the conventional animals survive longer term on this artificially high fat diet? Only a 6 hour time point is shown. Is the EEC desensitization process reversed? How does this acute fat intake relate to the mammalian diet since this could have more acute toxicity effects. The dietary fat composition used should be better justified and explained in this context.

The reviewer raises an important question about whether the EEC that undergo silencing in the presence of microbiota are viable and recover to normal function later in the post-prandial cycle. We agree that this is an important point and acknowledge that most of our data focus on the onset of EEC silencing without a fully complementary analysis of the recovery process. To help address this issue, we offered to conduct the following experiment:

(i) In Figure 4E of the submitted manuscript, we performed a temporal tracing of EEC morphology following different HF treatment time (4hours - 10hours). In order to address the question whether EEC desensitization is reversible or not, we could perform the 6hours HF treatment on zebrafish and then let the them recover in fresh media for longer periods of time, then measure EEC morphology and EEC function response.

Lipid digestion, absorption and metabolism in the intestine is well conserved between zebrafish and mammals. We adopted the egg yolk diet as our HF feeding paradigm because chicken egg yolk is the most used HF diet in zebrafish larvae and juveniles for metabolism and obesity research (PMIDs: 30177968, 22721970, 21724975, 25497901, 26607039). We believe the egg yolk diet is relevant because it is a rich source of dietary lipid (60% lipid by dry weight) and is a common dietary constituent for humans and other animals. We agree with the reviewer that our feeding paradigm does not recapitulate a regular mammalian diet, but this was not our goal. The fact that the lipase inhibitor Orlistat reverses HF diet-induced EEC silencing leads us to believe the phenotype is mediated by lipid digestion and processing, both of which are important in mammalian gut physiology. To what extent the dynamic responses we observed here are conserved in mammals is unknown, but we believe these studies introduce new concepts for consideration. In response to the reviewer’s concerns, we have now added text to the subsection “EEC physiology in zebrafish” and subsection “High fat feeding” subsectionclarifying why this particular HF meal model was chosen, the potential roles of specific lipid and other components, and how those nutrient levels compare to commonly used rodent HF diets.

3) It is not clear that the ER stress response is functionally coupled to EEC hormone physiology in this model. The ER stress response may reflect that experiments were done in GCaMP6, rather than wildtype animals. GCaMP proteins buffer intracellular calcium levels and it is possible that EEC could be more vulnerable to ER stress in this context and lipid exposure in this case is a second hit.

We thank the reviewer for raising this concern and we agree that these side effects of Gcamp expression have been reported in mouse studies. We agree that this should be taken into account for the Gcamp fluorescence data in our studies. However, the experiments in which HF diet induced changes in EEC morphology were not performed in Gcamp6 transgenic zebrafish (Figure 4, Figure 6J-K and O-Q, Figure 7G-P, Figure 8A-B andI, Figure 9H and Figure 4—figure supplement 1A-C, Figure 4—figure supplement 4C-D, Figure 4—figure supplement5 and Figure 9—figure supplement 3A-D in current manuscript). In addition, the experiment demonstrating an increase of s-xbp1 and other UPR transcripts following HF feeding was also not performed in Gcamp6 transgenic zebrafish (Figure 6L-N, Figure 7I,J,O and Figure 8J). Therefore, we think our evidence supports that Gcamp toxicity is not responsible for the phenotype that we observed in our study. If the reviewer thinks more information here is critical, we could do the following experiment:

(j) We could compare the basal ER stress and UPR markers in Gcamp6+ transgenic fish and Gcamp6- wildtype fish following HF feeding using qRT-PCR.

4) In the figures and videos, the high fat diet appears to trigger a distended GI that could be involved in the EEC silencing. Since SGLT-1 is a known cell volume regulator and inhibits the EEC signaling in the same manner, some data to rule out abnormal cell volume regulation in this model seems important. This is highly relevant in the context of store-operated (Orai-STIM) and/or TRP channels which couple apical plasma membrane to the ER and are primary regulators of Ca^2+^ transients associated with cell volume regulation.

We agree that the intestinal lumen is indeed distended in animals that have recently consumed the HF meal, a natural consequence of meal ingestion. It’s important to note here that we showed that consumption of a standard zebrafish diet was insufficient to induce EEC silencing (Figure 7—figure supplement 1D), so we are doubtful that the physical distension itself is driving the EEC silencing phenotype. The reviewer raises the question of whether the observed change in EEC morphology as a part of the EEC silencing phenotype may be due to effects on signal transduction pathways controlling cell volume, presumably in EECs but perhaps other intestinal cell types. We agree that regulated changes in EEC cell volume might be contributing to the EEC morphology phenotype, but have completed a new experiment below that tends to exclude this as a possibility. In the present study, we have shown that feeding palmitate fatty acid alone is sufficient to cause fatty acid insensitivity (Figure 7—figure supplement 1D) without inducing the EEC morphological changes caused by the complete HF diet (Figure 7—figure supplement 1A-C). This result indicates that changing EEC morphology alone is not sufficient to drive the full phenotype of EEC silencing. For all these reasons, we present the EEC silencing phenotype as having two components -nutrient insensitivity and morphological change. All that being said, if it is important to the reviewer that we quantify changes in cell volume or test the pathways they suggested, we could perform the following experiments:

(k) We could measure EEC cell volume at different HF treatment timepoints to see whether there is a cell volume change in response to HF feeding.

(l) If cell volume changes are detected, we could test whether Orai-Stim inhibitor MRS 1845 (PMID: 12527326, PMID: 19696927, PMID: 18755743) is able to reverse the phenotype.

(m) If cell volume changes are detected, we could use glibenclamide (an inhibitor for SUR1-TRPM4 which is shown to mediate astrocyte cellular volume change during stroke, PMID: 24380477) to test whether the phenotype can be reversed.

The data from (k) are displayed in Figure 4—figure supplement 3D. These data are explained in the subsection” High fat feeding induces morphological adaption in enteroendocrine cells”. This new analysis of EEC cell volume following HF feeding and morphologic changes associated with EEC silencing did not reveal significant changes of EEC cell volume. This suggests cell volume regulation is not a major factor in EEC silencing.

As an aside, we also considered that the changes in EEC morphology may be caused indirectly by the accumulation of numerous large cytosolic lipid droplets in absorptive enterocytes that surround each EEC within the epithelium (see PMID 27655916). Increased enterocyte size due to any of these upstream processes could “squeeze” EECs into an apparent closed morphological state. Our data showing blocking lipase activity with Orlistat prevents EEC silencing (Figure 7) is consistent with this model too. However, our previous study showed that animals treated the same way (given the same HF meal without prior feeding) accumulate a relatively modest volume of enterocyte lipid droplets, compared to fish given the same HF meal after having been provided a normal diet for several days before (in PMID 22980325, compare “starved” with “C-fed” in Figure 2). This suggests that enterocytes in the present study are not accumulating maximal enterocyte lipid droplet volume, so we think it’s unlikely that limited accumulation of enterocyte lipid droplets would be sufficient to physically impose our observed changes on EEC morphology after HF feeding. In the long term, we are interested in further testing this idea, but we unfortunately don’t have clean ways of suppressing enterocyte lipid droplet accumulation in zebrafish (see our responses to reviewer 3’s concern #4 below), which is why we didn’t attempt that in the paper.

Reviewer #3:[…] Overall, the present study is novel, well executed and elegantly presented. All major morphological observations were quantified appropriately. Raw data together with results from the performed statistical testing were made publicly available by the authors to support reproducibility.Essential revisions:1) Given the observed silencing of EECs after HFD, the finding of increased transcription of several genes encoding for EEC hormones (Figure 5A) is counter-intuitive and suggests a translational block and/or impairment of vesicle excretion. It would be of value to discuss this.

We thank the reviewer for raising this point, and we apologize for not providing a clearer interpretation. In the initial manuscript, we hypothesized that the increase in EEC hormone expression is an early post-prandial step (i.e., perhaps 1-3 hours after consumption of HF diet), whereas the EEC silencing phenotypes such as UPR induction occur later in the post-prandial process (i.e., by 6 hours after HF feeding). However, most of our data focus on a 6 hour timepoint, so we didn’t know how early EEC hormones are induced, and how related temporally to UPR induction. To help test our hypothesis and resolve this issue, we proposed to do the following new experiment:

(n) We could perform a qRT-PCR experiment to measure EEC hormone changes at different timepoints after HF feeding (e.g., 4hours, 6hours, 8hours, 10hours). If ER stress induces translational block as the reviewer predicts, we might expect to see a transient increase of EEC hormone transcript that may later decrease. We will also further discuss this in our text to ensure early vs late postprandial steps are clearly presented.

We have now performed the indicated experiment (n) above and the data are displayed in Figure 6—figure supplement 1A-F. These data are further explained in subsection “Activation of ER stress following high fat feeding leads to EEC silencing”. We observed that ER stress/UPR genes (bip and grp94) are induced as early as 2 hours after HF feeding and stay elevated through 10 hours. This is coincident with the induction of the hormone gene *ccka* transcript by 2 hours after HF feeding which plateaued afterwards. Whereas the hormone gene *pyyb* transcript continued to elevate until 6 hours after HF feeding and plateaued afterwards. Interestingly, the EEC silencing phenotype starts to manifest around 6 hours after HF feeding when the hormone gene transcription response plateaued. As suggested in our working model, our data support the hypothesis that at early postprandial stage, EEC increases hormone transcription and secretion to compensate for the high fat challenge. However, during late postprandial stage (>6 hours HF feeding), EECs adapt into a silent state. We posit that continuous ER stress which is induced early and throughout the HF feeding underlies the gradual appearance of EEC silencing later 6h after HF feeding. In addition, our data suggest that HF feeding reduces PYY hormone protein content in EECs despite the transcript increase. The reduction of PYY hormone protein may be either due to depletion of the protein content through increased hormone secretion, or due to the translation block from the elevated ER stress as suggested by the reviewers. We also added more text in the subsection “EEC silencing” about the relationship between ER stress, early and late postprandial response in the manuscript. We thank the reviewer for suggesting this experiment, as it has helped clarify our working model.

2) The absence of a significant increase of Xbp1t and Xbp1s after HFD (Figure 5B) is surprising in light of the observed increase in Grp78 (Bip) and the reported findings from the Xbp1s reporter model. This discrepancy might be due to a dilution effect, as whole tissue was analyzed in Figure 5B. Although not critical, it would be interesting to see the results of Xbp1 splicing assays of these samples (i.e. electrophoresis of amplified Xbp1 transcripts).

We agree with the reviewer that the discrepancy might be due to a dilution effect. It is important to point out that our data with the ef1a:xbp1d-GFP reporter line in Figure 4JK, which expresses GFP only when xbp1 transcript is spliced, shows enrichment in EECs after HF feeding, suggesting that there is indeed elevated xbp1 splicing in those cells.

Below we list an experiment we could do to further address the reviewer’s concern, but it would be very challenging. Given that this point is non-critical and the large number of other additional experiments, we would prefer not to undertake this experiment at this time.

(o) Nevertheless, we could perform FACS to sort EECs from zebrafish larval intestines and perform qRT-PCR for these genes. We could attempt to resolve xbp1 transcripts by electrophoresis. However, this is challenging considering the small size of zebrafish larvae and low number of EEC/animal.

3) Thapsigargin induces ER stress via specific inhibition of sarco-endoplasmic reticulum Ca^2+^-ATPases, which subsequently raises cytosolic calcium. This is an important confounding variable when using a calcium reporter, as it is done here. It would be critical to provide raw fluorescence intensities for Figure 5H and I (as done in Figure 2D) as the reporter signal may already be saturated at baseline. Alternatively, the authors could provide functional data for the BFA or TUDCA experiments as these compounds should not affect calcium signaling.

We agree with the reviewer about the caveats of using thapsigargan with a calcium reporter like Gcamp. This concern motivated our decision to repeat those experiments with a second ER stress inducer, Brefeldin A, which showed similar effects on EEC morphology as thapsigargan (Figure 6J-K). However, we agree this deserves more attention, so we had provided the raw fluorescence data for Figure 5H and I (Figure 6H-I in the current manuscript) in the new figure 9—figure supplement 1, and could perform the following new experiment:

(p) We could treat zebrafish with BFA and TUDCA to test their Gcamp responses to palmitate and/or glucose, as we did for thapsigargan in Figure 5.

4) Pre-treatment of animals with palmitate (instead of HFD) reduced the ability of EECs to respond to subsequent palmitate stimuli but did not induce the morphological changes seen after HFD (Figure 7—figure supplement 1). This unexpected finding, which suggests the presence of at least two independent mechanisms of EEC silencing, is attributed to "undefined signals from fat digestion" (subsection “The effects of diet and microbes on EEC silencing”), which is too vague in my eyes. The suggested role of mechanical pressure on EECs by adjacent enterocytes that have accumulated lipid droplets is not supported by experimental data. Is glycerol thought to play a role? Is the length/nature of fatty acids important? Does the palmitate diet increase EEC ER stress? It would valuable to at least discuss the former two points and to experimentally address the latter one.

See also our response to reviewer 2’s point #4 above. We agree that it would be helpful to directly test the role of lipid droplet accumulation in enterocytes on EEC silencing, but we are unaware of a way to block lipid droplet formation that doesn’t deleteriously affect other aspects of lipid absorption. For this reason, we used the lipase inhibitor Orlistat and found that it blocks EEC silencing and presumably also impairs lipid droplet accumulation. We have tested mammalian DGAT inhibitors in zebrafish but they do not appear to be efficacious. Experiments in cultured cells and yeast have shown that exposing cells to fatty acids while blocking lipid droplet formation kills the cell (see PMIDs 19690167 and 24273168). We therefore are unable to test the requirement for lipid droplet biogenesis in this EEC silencing model, though we remain interested in this possibility. We have also not tested the ability of glycerol, different fatty acids, mono- or di-acylglyerols in this system. As the reviewer suggests, we have more thoroughly discussed these issues in the revised manuscript in the subsection “EEC silencing”. Also see our proposed experiment (l) above in our response to reviewer 2’s point #4, where we measured EEC volume after HF diet feeding (Figure 4—figure supplement 3D). To further address the reviewer’s concern, we proposed to do the following experiment:

(q) We will perform the requested experiment to test whether palmitate increases ER stress in a xpb1-GFP reporter line and qRT-PCR of ER stress related genes after HF diet feeding.

The data are displayed in Figure 7—figure supplement 1F-H. These data are further explained in the subsection “Blocking fat digestion and absorption inhibits EEC silencing following high fat feeding”. Briefly, these data show that palmitate treatment was not sufficient to induce significant ER stress activation like HF feeding does. In summary, these data support the reviewer’s comment and our conclusion that there are multiple independent mechanisms that can mediate EEC silencing and we have added more thorough discussion in the revised manuscript (subsection “EEC silencing”).

5) The authors convincingly demonstrate that mono-colonization of animals with Acinetobacter sp. ZOR0008 is sufficient to permit HFD-induced changes of EEC morphology. It remains unclear, however, whether this observation is accompanied with a reduced EEC response to fatty acids and/or glucose. This is a challenging experiment as it would require the generation of germ-free neurod1:Gcamp6f fish and is not absolutely critical. Nonetheless – if feasible – these experiments would significantly strengthen the manuscript given the observation that morphologic and functional EEC silencing can be uncoupled in certain conditions as described in point 4.

We agree with the reviewer’s comments and we proposed to perform the requested experiment:

(r) We will monoassociate germ-free zebrafish with Acinetobacter and control bacterial strains, and test EEC responsiveness to palmitate and/or glucose after a HF meal.

The data are displayed in the Figure 9—figure supplement 2. These data are explained in the subsection “High fat feeding induces EEC silencing in a microbiota dependent manner”. Our data show that mono-association of *Acinetobacter* sp. significantly reduces EECs’ nutrient response following HF feeding, similar to complex microbiota.